# Q-Supervised Contrastive Representation: A State Decoupling Framework for Safe Offline Reinforcement Learning

Zhihe Yang[1]  Yunjian Xu[1]  Yang Zhang[1 2]

## Abstract

Safe offline reinforcement learning (RL), which aims to learn the safety-guaranteed policy without risky online interaction with environments, has attracted growing recent attention for safety-critical scenarios. However, existing approaches encounter out-of-distribution problems during the testing phase, which can result in potentially unsafe outcomes. This issue arises due to the infinite possible combinations of reward-related and cost-related states. In this work, we propose *State Decoupling with Q-supervised Contrastive representation* (SDQC), a novel framework that decouples the global observations into reward- and cost-related representations for decision-making, thereby improving the generalization capability for unfamiliar global observations. Compared with the classical representation learning methods, which typically require model-based estimation (e.g., bisimulation), we theoretically prove that our Q-supervised method generates a coarser representation while preserving the optimal policy, resulting in improved generalization performance. Experiments on DSRL benchmark provide compelling evidence that SDQC surpasses other baseline algorithms, especially for its exceptional ability to achieve almost zero violations in more than half of the tasks. Further, we demonstrate that SDQC possesses superior generalization ability when confronted with unseen environments.

## 1. Introduction

Reinforcement learning (RL) has been proven to be a powerful tool for solving high-dimensional decision-making

[1]Department of Mechanical and Automation Engineering, The Chinese University of Hong Kong, Hong Kong SAR, China. [2]Department of Electrical and Electronic Engineering, Hong Kong Polytechnic University, Hong Kong SAR, China. Correspondence to: Yunjian Xu <yjxu@mae.cuhk.edu.hk>.

*Proceedings of the $42^{nd}$ International Conference on Machine Learning*, Vancouver, Canada. PMLR 267, 2025. Copyright 2025 by the author(s).

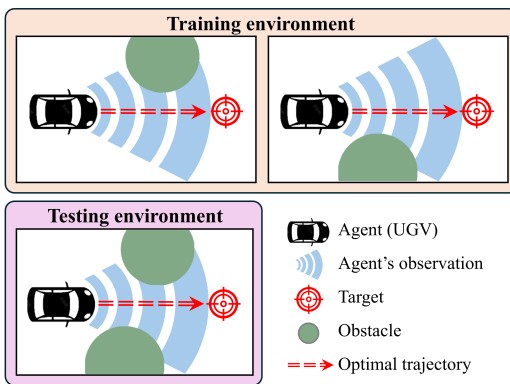

*Figure 1.* OOD issue for offline trained UGV in the testing phase.

problems under uncertainty (Mnih et al., 2015; Silver et al., 2017; Schrittwieser et al., 2020). Nevertheless, safety concerns remain a significant obstacle to the extensive adoption of RL in safety-critical domains (García & Fernández, 2015; Gu et al., 2022; Xu et al., 2022b; Li, 2023), such as industrial management, and robot control. In these contexts, the potential for catastrophic outcomes necessitates a significant emphasis on preventing unsafe actions (Andersen et al., 2020; Brunke et al., 2022). As a promising method that received growing attention, safe RL provides safety guarantees by formulating the problem as a constrained Markov decision process (CMDP) (Altman, 1998; 2021).

Over the past few years, a multitude of safe RL algorithms have been introduced (Achiam et al., 2017; Tessler et al., 2018; Zhao et al., 2021; Sootla et al., 2022; Yu et al., 2022). Regrettably, most existing methods address safety concerns within online settings, relying on the high-fidelity simulators or agent-environment interactions during the training process, which introduces additional risks of safety violations (Liu et al., 2023a). Safe offline RL, on the other hand, provides a promising solution that learns the safety-guaranteed policy in a fully offline manner. Its training requires no risky interaction with the environment and relies only on the pre-collected offline dataset. However, empirical observations indicate that most existing safe offline RL algorithms fail to thoroughly ensure pre-defined safety constraints during testing (Liu et al., 2023a; Zheng et al., 2024). Such occurrences tend to be more pronounced in environments characterized by higher observation dimensions.

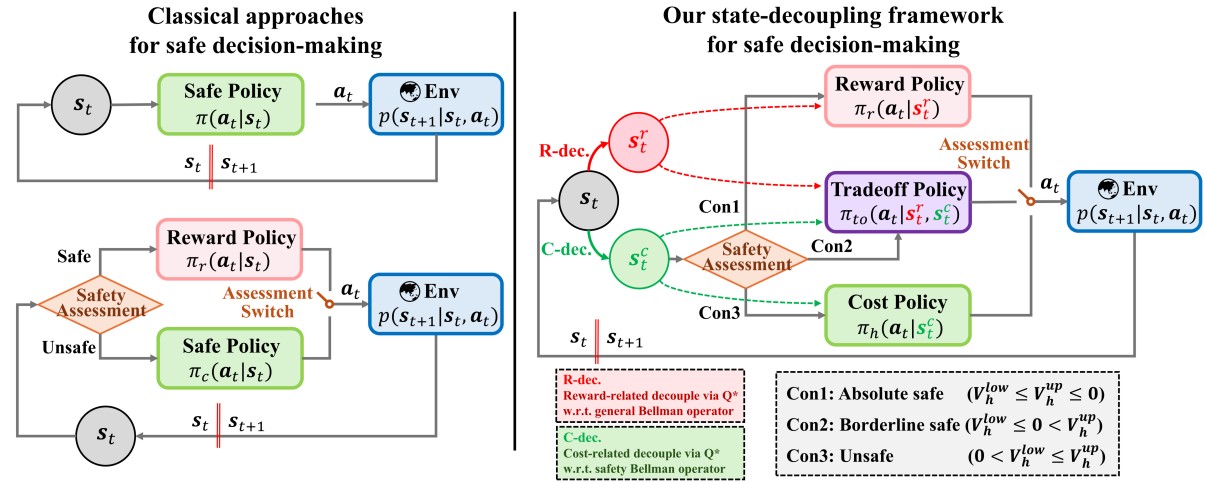

*Figure 2.* Overview diagram of classical approaches (left) and our proposed state-decoupling framework (right) for safe decision-making.

In offline RL, it is imperative that the states visited during testing have been included in (or at least not far away from) the training dataset to ensure robust performance (Fujimoto et al., 2019; Wang et al., 2022). However, Safe offline RL problems have various combinations of reward-related and cost-related states. For instance, as illustrated in Figure 1, a UGV (unmanned ground vehicle) needs to navigate around traps to reach its final destination. During testing, if the relative positions of traps and the target haven't occurred in the training dataset, the agent may struggle to make informed decisions based on such unknown observations. It is reasonable to suspect that the primary reason for the subpar performance of safe offline RL during tests lies in the out-of-distribution (OOD) issue.

To tackle this problem and improve the generalization of safe offline RL, we propose *State Decoupling with Q-supervised Contrastive representation* (SDQC), a novel representation learning method that decouples the global observations into reward- and cost-related representations. Attributable to the successful application of Hamilton-Jacobi (HJ) reachability analysis in Safe RL, which introduces a safety analysis method iterated through Q-learning with convergence guarantees, our approach makes safety assessments on the cost-related representations and make decisions based on the assessment results. Unlike classical methods (as depicted in the left subplot of Figure 2), which rely on global observations for decision-making, our SDQC is the **first** to utilize decoupled representations for decision-making in safe RL tasks (see the right subplot of Figure 2). It employs reward-related representations to make decisions when the assessment confirms absolute safety, switches to cost-related representations when the assessment deems the situation unsafe, and integrates both when the assessment indicates borderline safety.

Nevertheless, effective differentiation between reward- and cost-related information from global observations poses a formidable challenge, especially when certain dimensions of the observations contain intertwined information. For instance, some information, like speed and acceleration of UGV, should be included in both reward- and cost-related representations. On the other hand, information from the environment-detecting sensors, which include positions of destinations and obstacles, should be distinctly decoupled. Manual separation proves impractical in most cases.

Towards this end, our Q-supervised contrastive representation decouples the global observations through clustering representations that demonstrate similar learned-Q* across the actions in support[§]. The representations solely capture either reward or cost information, independent of another factor, as determined by the training of Q*. Unlike model-based representations learning (e.g., bisimulation), our SDQC circumvents model estimation, thus mitigating the challenges posed by severe estimation errors in scenarios with sparse rewards or costs. Moreover, we demonstrate that our representations can be trained concurrently with the Q*-learning process by incorporating an additional loss term within the framework of implicit Q-learning.

Further, we provide theoretical evidence that our method produces a coarser representation compared to bisimulation, while still preserving the optimal policy. This is supported by our argument that SDQC leads to a higher information entropy of the global observations when conditioned on the representations. This attribute grants SDQC superior generalization capabilities, bolstering its efficacy in handling OOD observations during the testing phase.

---

[§]For simplicity, we denote $Q^*$ as a generic notaiton to represent the optimal Q-value functions for both reward ($Q_r^*$) and cost ($Q_h^*$) in this paper. Similarly, $Q$ refers to both $Q_r$ and $Q_h$.

The experimental results showcase that our SDQC outperforms other safe offline RL algorithms in the DSRL benchmark, especially in its exceptional ability to achieve zero violations in the majority of tasks. Further, in generalization tests where agents are evaluated in environments that differ from the training ones, all baseline algorithms show a substantial increase in cost and/or a significant decline in reward. In contrast, SDQC stands out as the only approach that guarantees no increase in cost while experiencing only a slight decay in reward.

## 2. Preliminaries

**Safe Offline RL.** Safe RL tasks are generally modeled as CMDP in the form of $\mathcal{M} = (\mathcal{S}, \mathcal{A}, P, r, c, \gamma, d_0)$, where $\mathcal{S}$ is the state space, $\mathcal{A}$ is the action space, $P$ is the model dynamics, $r : \mathcal{S} \times \mathcal{A} \to \mathbb{R}$ represents the reward function, $c : \mathcal{S} \times \mathcal{A} \to [0, C_{max}]$ represents the cost function, $\gamma \in [0, 1)$ is the discount factor, and $d_0 \in \Delta(\mathcal{S})$ is the distribution of initial state $s_0$ (the set of the probability distribution over $\mathcal{S}$ is denoted as $\Delta(\mathcal{S})$). $P(s'|s, a) : \mathcal{S} \times \mathcal{A} \to \Delta(\mathcal{S})$ represents the transition function from state $s$ to $s'$ when taking action $a$. The state-action-reward-cost transitions over trajectory are recorded as $\tau := (s_t, a_t, r_t, c_t)_{t \geq 0}$. The goal of Safe RL is to learn a policy $\pi : \mathcal{S} \to \Delta(\mathcal{A})$ that maximizes the expectation of the cumulated discounted reward while restricting the expected cumulative costs below a predefined cost limit $\kappa$, which can be denoted by $\max_\pi \mathbb{E}_{\tau \sim \pi}[\sum_{t=0}^\infty \gamma^t r(s_t, a_t)]$, s.t. $\mathbb{E}_{\tau \sim \pi}[\sum_{t=0}^\infty \gamma^t c(s_t, a_t)] \leq \kappa$.

In offline settings, the training is performed on a statistical dataset denoted as $\mathcal{D}_\beta := (s, a, s', r, c)$. This offline dataset comprises both safe and unsafe trajectories and is acquired from a behavior policy $\pi_\beta$. During training, most existing safe offline RL algorithms utilize the temporal difference (TD) method to learn the reward state-value function $V_r^\pi(s_t)$, which models the expected cumulative reward $\mathbb{E}_{\tau \sim \pi}[\sum_{i=t}^\infty \gamma^i r(s_i, a_i)]$, as well as the cost state-value function $V_c^\pi(s_t)$, which models the expected cumulative cost $\mathbb{E}_{\tau \sim \pi}[\sum_{i=t}^\infty \gamma^i c(s_i, a_i)]$. The primal training objective of safe offline RL can be expressed as follows:

$$\max_\pi \mathbb{E}_{s_t \sim \mathcal{D}_\beta}[V_r^\pi(s_t)]$$
$$\text{s.t. } \mathbb{E}_{s_t \sim \mathcal{D}_\beta}[V_c^\pi(s_t)] \leq \kappa; \ D(\pi|\pi_\beta) \leq \epsilon_\pi, \tag{1}$$

where $D(\pi|\pi_\beta)$ is the divergence term that prevents the distributional shift in offline training.

A commonly employed approach for solving Eq. (1) involves reformulating the training objective using the Lagrangian dual form as $\min_{\lambda \geq 0} \max_\pi \mathbb{E}_{s_t \sim \mathcal{D}_\beta}[V_r^\pi(s_t) - \lambda(V_c^\pi(s_t) - \kappa)]$, s.t. $D(\pi|\pi_\beta) \leq \epsilon_\pi$, where the learnable Lagrange multiplier $\lambda$ is iteratively updated to enforce the constraint. However, the Lagrangian approach can be sensitive to the learning rate and initialization of the multiplier

(Stooke et al., 2020). Furthermore, the joint optimization of $V_r^\pi$, $V_c^\pi$, and $\pi$ leads to significant instability, as minor approximation errors can bootstrap across them and propagate, thereby undermining the ability to provide robust safety guarantees (Kumar et al., 2019; Zheng et al., 2024).

**Hamilton-Jacobi reachability.** As a prospective method to perform safety assessment rooted in control theory, HJ reachability has been proven to be applicable in Safe RL tasks for both online settings (Chen et al., 2021a; Yu et al., 2022) and offline settings (Zheng et al., 2024). In addition to the tuple formulation $\mathcal{M}$ in CMDP, we introduce a constraint violation function $h : \mathcal{S} \to \mathbb{R}$, which is positive if the state constraint is violated and negative otherwise. For a given state $s$, the safe value function $V_h^\pi(s) := \max_{t \in \mathbb{N}}\{h(s_t) \mid s_0 = s, a_i \sim \pi(\cdot|s_i), \forall i \in \{0, \dots, t\}\}$ represents the worst constraint violations among all possible trajectories induced by policy $\pi$. The corresponding safe Q-function is given by $Q_h^\pi(s, a) := \max_{t \in \mathbb{N}}\{h(s_t) \mid s_0 = s, a_0 = a, a_i \sim \pi(\cdot|s_i), \forall i \in \{1, \dots, t\}\}$. The optimal safe value function, defined as $V_h^*(s) := \min_\pi V_h^\pi(s)$, stands for the smallest violation one can obtain through adjusting the policy $\pi$. Similarly, the corresponding optimal safe Q-function can be expressed as $Q_h^*(s, a) := \min_\pi Q_h^\pi(s, a)$. With the discount factor defined as $\gamma$, Fisac et al. (2019) introduce the following safety Bellman operators:

$$\mathcal{B}_h^* Q_{h,\gamma}(s, a) := (1 - \gamma)h(s) + \gamma \max\{h(s), V_{h,\gamma}(s')\},$$
$$V_{h,\gamma}(s') = \min_{a'} Q_h(s', a'), \tag{2}$$

which is a contraction mapping satisfying $\lim_{\gamma \to 1} Q_{h,\gamma} \to Q_h^*$, $\lim_{\gamma \to 1} V_{h,\gamma} \to V_h^*$. A direct safety inference can be made after training converges. $V_h^*(s) \leq 0$ implies the existence of policies that guarantee adherence to the hard constraints throughout the trajectory. Conversely, $V_h^*(s) > 0$ indicates that the destiny towards unsafe states regardless of the chosen policy. In the offline settings, Zheng et al. (2024) pioneered the application of HJ reachability analysis for safety assessment. They present that the decision-making for Safe RL with hard constraints can be decoupled as:

$$\begin{cases} \textbf{Safe} : \max_\pi \mathbb{E}_{a \sim \pi}\left[A_r^*(s, a) \cdot \mathbb{I}_{V_h^*(s) \leq 0}\right] \\ \quad \text{s.t. } \int_{\{a|Q_h^*(s,a) \leq 0\}} \pi(a|s)da = 1; \ D(\pi|\pi_\beta) \leq \epsilon \\ \textbf{Unsafe} : \max_\pi \mathbb{E}_{a \sim \pi}\left[-A_h^*(s, a) \cdot \mathbb{I}_{V_h^*(s) > 0}\right] \\ \quad \text{s.t. } \int_a \pi(a|s)da = 1; \ D(\pi|\pi_\beta) \leq \epsilon \end{cases} \tag{3}$$

where $A_r^*(s, a) := Q_r^*(s, a) - V_r^*(s)$ and $A_h^*(s, a) := Q_h^*(s, a) - V_h^*(s)$. Eq. (3) theoretically ensures zero constraint violations. However, challenges arise from estimation errors and OOD problems during the testing phase. As a result, the empirical results demonstrate the inability of their algorithm (FISOR) to achieve absolute safety guarantees.

# 3. State Decoupling with Q-supervised Contrastive Representations

In our state-decoupling framework, we aim to decouple the state $s$ into two separate representations: one related to rewards, denoted as $s_r$, and the other related to costs, denoted as $s_c$. We slightly abuse the notation $z_\theta(s)$ (or simplified as $z$) to represent the neural network embedded representations of either $s_r$ or $s_c$ in this section.

## 3.1. Motivation

Manually abstracting representations of reward-related or cost-related aspects directly from original state observations can be challenging due to the entanglement of information within certain observation dimensions. It is observed that the optimal Q-value, whether associated with reward or cost, exclusively encompasses the information it was trained with, independent of another factor. For instance, concerning the states of the agent depicted in Figure 1, the optimal Q-values related to reward are the same across all actions regardless of the cost-related observations. These states should be embedded as the same reward-related representation. To achieve a coarser abstraction while maintaining the optimal Q-value unchanged, we design the objective for both reward- and cost-related representations as follows

$$\max_\theta H(s|z_\theta(s))$$
$$\text{s.t. } (\mathcal{B}^* Q(z_\theta(s), a) - Q(z_\theta(s), a))^2 \leq \epsilon_\mathcal{B}, \ \forall a \in \mathcal{A}, \tag{4}$$

where $H(\cdot|\cdot)$ represents conditional entropy (Shannon, 1948), $\epsilon_\mathcal{B}$ is an arbitrary small number and $\mathcal{B}^*$ is the optimal general/safety Bellman operator. We define $d(s_1, s_2) := \sup_{a \in \mathcal{A}} |Q^*(z_\theta(s_1), a), Q^*(z_\theta(s_2), a)|$ as the distance measure between a pair of states $s_1, s_2 \in \mathcal{S}$. One can always find an arbitrarily small number $\epsilon_d$ such that the objective in Eq. (4) can be achieved through embedding the states $\mathcal{C}(s') := \{\tilde{s} \in \mathcal{S} \mid d(\tilde{s}, s') < \epsilon_d\}$ with the same representation for any $s' \in \mathcal{S}'$, where $\mathcal{S}'$ is a smallest subset of $\mathcal{S}$ such that for any $s_1', s_2' \in \mathcal{S}'$, we have $d(s_1', s_2') \geq 2\epsilon_d$ and $\bigcup_{s' \in \mathcal{S}'} \mathcal{C}(s') = \mathcal{S}$. Contrastive learning, which aims to bring artificially defined similar instances closer and push other instances further apart in the representation space (Oord et al., 2018; Bachman et al., 2019; Chen et al., 2020), provides a promising solution for our embedding task.

## 3.2. Q-supervised Contrastive Representation

Inspired by Agarwal et al. (2021), we adopt a soft similarity measure, denoted as $\Gamma(s, \tilde{s}) = \exp(-d(s, \tilde{s})/\eta)$, to quantify the distance between two states (with $\eta$ representing the temperature factor). Notably, directly calculating the distance measure involves querying out-of-distribution (OOD) actions in the offline setting. To address this issue, we pre-train a generative model to capture the behavior pol-

icy (cf. Appendix B.2,C.1 for details). This allows us to generate in-support actions for any given states in the offline dataset, denoted as $\mathcal{A}_\beta^s$. As a result, we have the approximation $d(s, \tilde{s}) \approx \sup_{a \in \mathcal{A}_\beta^s} |Q^*(z_\theta(s), a) - Q^*(z_\theta(\tilde{s}), a)|$ for calculating the soft similarity measure.

In practice, we employ a random sampling approach to select a subset of states, denoted as $\mathcal{S}'$, from the offline state set. Within the subset, we further randomly choose a set of anchor states, denoted as $\{s_i \in \mathcal{S}' \mid i \in \mathcal{I}\}$, where $\mathcal{I}$ represents the index set of the selected anchor states. For each anchor state $s_i$, we use its nearest neighbor in $\mathcal{S}'$ based on the similarity measure $\Gamma$ to define the positive pairs $\{s_i, \tilde{s}_i\}$, where $\tilde{s}_i = \arg\min_{s \in \mathcal{S}' \setminus \{s_i\}} \Gamma(s_i, s)$. The remaining states in $\mathcal{S}'$ are considered as negative samples. Attention-based or multiple-layer-perceptron-based neural networks are utilized to encode the state as a normalized vector on the unit hypersphere, i.e., $\|z_\theta(s)\| = 1$ (cf. Appendix C.2 for detailed network selection and structure design). Finally, we have the following contrastive loss, which encourages the embedding of states with similar $Q^*$ values across all actions to have similar representations:

$$\mathcal{L}_\theta = \sum_{i \in \mathcal{I}} -\frac{1}{|\mathcal{I}|} \log \frac{\text{POS}_i}{\text{POS}_i + \text{NEG}_i},$$
$$\begin{cases} \text{POS}_i = \Gamma(s_i, \tilde{s}_i) \exp(z_i \cdot \tilde{z}_i / \nu) \\ \text{NEG}_i = \sum_{z_j \in \mathcal{Z}' \setminus \{z_i, \tilde{z}_i\}} (1 - \Gamma(z_i, z_j)) \exp(z_i \cdot z_j / \nu) \end{cases} \tag{5}$$

where $\nu$ is a temperature parameter. It is important to note that Eq. (5) requires precise calculation of optimal Q-values for all states across all actions, i.e., the constraints in Eq. (4) are satisfied. However, the Q-values are derived from the representation network, and even small changes in the network can result in variations in the Q-values. Therefore, it is necessary to integrate the training process of the representation with the Q-learning process. Such coupling ensures that both the representation and Q-values are jointly updated, accounting for the interdependencies between them.

## 3.3. Practical Implementation

Building upon in-sample learning methods (Kostrikov et al., 2021; Xu et al., 2023; Garg et al., 2023; Zheng et al., 2024), our approach follows a two-step process. In the initial phase, we undertake the learning process for the value functions and representations associated with cost and reward separately. Following that, we extract the policy based on the acquired value functions and representations.

**Reward-related Representation.** We use implicit Q-learning (IQL) (Kostrikov et al., 2021) (cf. Appendix B.1 for details) to approximate the reward-related optimal value functions $Q_r^*$ and $V_r^*$ within the support of data distribution through upper expectile regression:

$$\mathcal{L}_{V_r} = \mathbb{E}_{(s,a)\sim D_\beta}\big[L_{\text{up}}^\tau(Q_r(z_{\theta_r}(s),a) - V_r(z_{\theta_r}(s)))\big], \quad (6)$$

$$\mathcal{L}_{Q_r} = \mathbb{E}_{(s,a,s',r)\sim D_\beta}\big[(r + \gamma V_r(z_{\theta_r}(s')) - Q_r(z_{\theta_r}(s),a))^2\big] \quad (7)$$

where $L_{\text{up}}^\tau(u) = |\tau - \mathbb{I}(u < 0)|u^2$, $\tau \in (0.5, 1)$. The reward-related representations are learned with the addition of a contrastive loss term (Eq. (5)), with a weighting factor of $\delta$. Consequently, the overall loss for the reward-related value functions and representations is formulated as:

$$\mathcal{L}_{reward} = \mathcal{L}_{V_r} + \mathcal{L}_{Q_r} + \delta\mathcal{L}_{\theta_r}. \quad (8)$$

**Cost-related Representation.** Similar to Zheng et al. (2024), we employ the safety Bellman operator, as denoted in Eq. (2), and utilize lower expectile regression to learn the cost-related optimal value functions $Q_h^*$ and $V_h^*$:

$$\mathcal{L}_{V_h^{\text{low}}} = \mathbb{E}_{(s,a)\sim D_\beta}\Big[L_{\text{low}}^\tau(Q_h(z_{\theta_h}(s),a) - V_h^{\text{low}}(z_{\theta_h}(s)))\Big], \quad (9)$$

$$\mathcal{L}_{Q_h} = \mathbb{E}_{(s,a,s',h)\sim D_\beta}\big[((1-\gamma)h + \gamma\max\{h, V_h^{\text{low}}(z_{\theta_h}(s')\} \\ - Q_h(z_{\theta_h}(s),a))^2\big], \quad (10)$$

where $L_{\text{low}}^\tau(u) = |\tau - \mathbb{I}(u > 0)|u^2$, $\tau \in (0.5, 1)$. Additionally, we learn an upper-bound cost-related value function $V_h^{\text{up}}$ to model the maximum $Q_h^*$ across all actions in support:

$$\mathcal{L}_{V_h^{\text{up}}} = \mathbb{E}_{(s,a)\sim D_\beta}\big[L_{\text{up}}^\tau(Q_h(z_{\theta_h}(s),a) - V_h^{\text{up}}(z_{\theta_h}(s)))\big]. \quad (11)$$

By incorporating an additional contrastive loss term (Eq. (5)) with a weighting factor of $\delta$, we express the overall loss for the cost-related value functions and representations as:

$$\mathcal{L}_{cost} = \mathcal{L}_{V_h^{\text{low}}} + \mathcal{L}_{V_h^{\text{up}}} + \mathcal{L}_{Q_h} + \delta\mathcal{L}_{\theta_h}. \quad (12)$$

**Policy Extraction.** As illustrated in the right subplot of Figure 2, we divide the global policy into three components: the reward policy $\pi_r$, which solely depends on the reward-related representation; the cost policy $\pi_h$, which solely relies on the cost-related representations; and the tradeoff policy $\pi_{to}$, which depends on both. We independently train the three policies using weighted regressed diffusion models, an approach pioneered by Zheng et al. (2024). They present that the optimal policy satisfies $\pi^*(a|z) \propto \pi_\beta(a|z) \cdot w(z, a)$, and the optimal policy can be obtained through weighted training of diffusion models. The weighted loss function for the three policies can be expressed as follows:

$$\begin{cases} \pi_r : \mathcal{L}_{\pi_r} = \mathbb{E}_{\text{var}}\big[w_r(z_{\theta_r}(s),a) \cdot \|\zeta - \zeta_{\psi_r}(a_t, z_{\theta_r}(s), t)\|\big] \\ \pi_h : \mathcal{L}_{\pi_h} = \mathbb{E}_{\text{var}}\big[w_h(z_{\theta_h}(s),a) \cdot \|\zeta - \zeta_{\psi_h}(a_t, z_{\theta_h}(s), t)\|\big] \\ \pi_{to} : \mathcal{L}_{\pi_{to}} = \mathbb{E}_{\text{var}}\big[w_{to}(z_{\theta_r}(s), z_{\theta_h}(s),a) \cdot \\ \qquad\qquad \|\zeta - \zeta_{\psi_{to}}(a_t, z_{\theta_r}(s), z_{\theta_h}(s), t)\|\big], \end{cases} \quad (13)$$

where var represents the variables involved in the expectation, with $t \sim U(1, T)$, $\zeta \sim \mathcal{N}(0, I)$, and $(s, a) \sim D_\beta$. The noised action $a_t = \alpha_t a + \sigma_t\zeta$ satisfies the forward transition distribution $\mathcal{N}(a_t|\alpha_t a, \sigma_t\mathbb{I})$ in the diffusion models, and $\alpha_t$, $\sigma_t$ are noised schedules. The weights in Eq. (13) are

$$\begin{cases} w_r(z_{\theta_r}(s), a) = \exp(\iota_r(Q_r(z_{\theta_r}(s), a) - V_r(z_{\theta_r}(s)))) \\ w_h(z_{\theta_h}(s), a) = \exp(-\iota_h(Q_h(z_{\theta_h}(s), a) - V_h(z_{\theta_h}(s)))) \\ w_{to}(z_{\theta_r}(s), z_{\theta_h}(s), a) = \exp(\iota_{to}(Q_r(z_{\theta_r}(s), a) - \\ \qquad V_r(z_{\theta_r}(s))) \cdot \mathbb{I}_{Q_h(z_{\theta_h}(s),a)\leq 0}, \end{cases} \quad (14)$$

where $\iota_r$, $\iota_h$ and $\iota_{to}$ are temperatures that control the behavior regularization strength.

After obtaining $\zeta_{\psi_r}$ $\zeta_{\psi_h}$ and $\zeta_{\psi_{to}}$, the three approximated optimal policies can be sampled through the reverse diffusion chain starting from random Gaussian noise (Ho et al., 2020; Song et al., 2020) (cf. Appendix B.3 for details). During the testing phase, we first perform safety assessments on the cost-related representations. If the assessment verifies absolute safety ($V_h^{\text{low}} \leq V_h^{\text{up}} \leq 0$), we employ the policy $\pi_r$. If the assessment indicates borderline safety ($V_h^{\text{low}} \leq 0 < V_h^{\text{up}}$), we utilize the policy $\pi_{to}$. In the case of an unsafe condition ($0 < V_h^{\text{low}} \leq V_h^{\text{up}}$), we rely on the policy $\pi_h$. See the right subplot of Figure 2 for details.

### 3.4. Comparison with Bisimulation

Bisimulation has been established as a useful tool for abstracting state representations (Definition 3.1), where states with identical transition and reward/cost functions are grouped together (Givan et al., 2003; Castro & Precup, 2010; Castro, 2020; Castro et al., 2021). However, employing bisimulation typically entails an additional step of training a model-based estimator to learn the state transition and reward/cost functions. Notably, the estimation of reward/cost functions becomes particularly challenging when the values are sparsely distributed (Lee et al., 2024). In contrast to such a model-based representation approach, learning the representations based on $Q^*$ (Definition 3.2) eliminates the necessity for estimating the exact model dynamics (Givan et al., 2003; Li et al., 2006). With $\Theta$ denoting a generic surjective mapping from the ground-truth state space to representation space, we have the following definitions:

**Definition 3.1.** *A bisimulation representation $\Theta_{\text{bisim}}$ is such that for any action $a$ and any represented state $z$, $\Theta_{\text{bisim}}(s_1) = \Theta_{\text{bisim}}(s_2)$ implies $r(s_1, a) = r(s_2, a)$ (or $c(s_1, a) = c(s_2, a)$) and $\sum_{s' \in \Theta_{\text{bisim}}^{-1}(z)} P(s'|s_1, a) = \sum_{s' \in \Theta_{\text{bisim}}^{-1}(z)} P(s'|s_2, a)$.*

**Definition 3.2.** *A $Q^*$-irrelevance representation $\Theta_{Q^*}$ is such that, $\Theta_{Q^*}(s_1) = \Theta_{Q^*}(s_2)$ implies $Q^*(s_1, a) = Q^*(s_2, a)$ for any action $a$.*

*Table 1.* Normalized DSRL benchmark results. The evaluation results are averaged over 3 random seeds (20 episodes for each seed). Gray: Unsafe agents. **Bold**: Safe agents whose normalized cost is smaller than 1. Red: Safe agents with the highest reward. Blue: Safe agents with the lowest cost.

| Task | BCQ-Lag | | CPQ | | COptiDICE | | CDT | | TREBI | | FISOR | | SDQC(ours) | |
|---|---|---|---|---|---|---|---|---|---|---|---|---|---|---|
| | reward↑ | cost↓ | reward↑ | cost↓ | reward↑ | cost↓ | reward↑ | cost↓ | reward↑ | cost↓ | reward↑ | cost↓ | reward↑ | cost↓ |
| *PointGoal1 | 0.71 | 4.29 | **0.56** | **0.93** | 0.40 | 5.53 | 0.21 | 1.59 | 0.36 | 2.79 | 0.68 | 4.19 | **0.35** | **0.36** |
| *PointGoal2 | 0.62 | 3.81 | 0.41 | 5.03 | 0.43 | 2.78 | 0.22 | 1.19 | 0.28 | 3.86 | 0.21 | 1.42 | **0.29** | **0.09** |
| *PointPush1 | 0.32 | 3.08 | 0.14 | 1.35 | 0.13 | 3.80 | 0.27 | 2.81 | 0.31 | 2.02 | 0.27 | 1.38 | **0.12** | **0.00** |
| *PointPush2 | 0.21 | 1.86 | 0.16 | 2.36 | 0.02 | 2.90 | 0.18 | 1.69 | 0.13 | 3.85 | 0.24 | 2.41 | **0.19** | **0.28** |
| *PointButton1 | 0.21 | 4.45 | 0.61 | 11.80 | 0.08 | 4.29 | 0.48 | 10.88 | 0.12 | 4.55 | **0.04** | **0.97** | **0.08** | **0.46** |
| *PointButton2 | 0.38 | 8.04 | 0.35 | 12.09 | 0.17 | 6.12 | 0.42 | 9.97 | 0.02 | 2.18 | 0.08 | 4.49 | **0.06** | **0.57** |
| CarGoal1 | 0.44 | 2.76 | 0.33 | 4.93 | 0.43 | 2.81 | 0.60 | 3.15 | 0.41 | 1.16 | **0.49** | **0.83** | **0.38** | **0.01** |
| CarGoal2 | 0.34 | 4.72 | 0.10 | 6.31 | 0.19 | 2.83 | 0.45 | 6.05 | 0.13 | 1.16 | **0.06** | **0.33** | **0.23** | **0.00** |
| CarPush1 | 0.23 | 1.33 | **0.08** | **0.77** | 0.21 | 1.28 | 0.27 | 2.12 | 0.26 | 1.03 | **0.28** | **0.28** | **0.30** | **0.00** |
| CarPush2 | 0.10 | 2.78 | -0.03 | 10.00 | 0.10 | 4.55 | 0.16 | 4.60 | 0.12 | 2.65 | **0.14** | **0.89** | **0.31** | **0.04** |
| CarButton1 | 0.13 | 6.68 | 0.22 | 40.06 | -0.16 | 4.63 | 0.17 | 7.05 | 0.07 | 3.75 | **-0.02** | **0.26** | **0.03** | 0.32 |
| CarButton2 | -0.04 | 4.43 | 0.08 | 19.03 | -0.17 | 3.40 | 0.23 | 12.87 | **-0.03** | **0.97** | **0.01** | **0.58** | **0.02** | 0.42 |
| AntVel | 0.85 | 18.54 | **-1.01** | **0.00** | 1.00 | 10.29 | **0.98** | **0.91** | 0.31 | **0.00** | 0.89 | **0.00** | 0.73 | **0.00** |
| HalfCheetahVel | 1.04 | 57.06 | 0.08 | 2.56 | **0.43** | **0.00** | **0.97** | 0.55 | 0.87 | 0.23 | 0.89 | **0.00** | 0.81 | **0.00** |
| SwimmerVel | 0.29 | 4.10 | 0.31 | 11.58 | 0.58 | 23.64 | 0.67 | 1.47 | 0.42 | 1.31 | -0.04 | **0.00** | -0.04 | **0.00** |
| **SafetyGym Average** | 0.39 | 9.17 | 0.16 | 9.05 | 0.26 | 5.66 | 0.46 | 4.93 | 0.25 | 2.10 | 0.28 | 1.25 | **0.26** | **0.17** |
| AntRun | 0.65 | 3.30 | **0.00** | **0.00** | 0.62 | 3.64 | 0.70 | 1.88 | 0.63 | 5.43 | **0.45** | **0.03** | 0.31 | **0.00** |
| BallRun | 0.43 | 6.25 | 0.85 | 13.67 | 0.55 | 11.32 | **0.32** | **0.45** | 0.29 | 4.24 | **0.18** | **0.00** | 0.20 | **0.00** |
| CarRun | 0.84 | 2.51 | 1.06 | 10.49 | **0.92** | **0.00** | 0.99 | 1.10 | 0.97 | 1.01 | 0.73 | 0.14 | 0.56 | **0.00** |
| DroneRun | 0.80 | 17.98 | 0.02 | 7.95 | 0.72 | 13.77 | **0.58** | **0.30** | 0.59 | 1.41 | 0.30 | 0.55 | 0.36 | 0.56 |
| AntCircle | 0.67 | 19.13 | **0.00** | **0.00** | 0.18 | 13.41 | 0.48 | 7.44 | 0.37 | 2.50 | **0.20** | **0.00** | **0.38** | **0.00** |
| BallCircle | 0.67 | 8.50 | 0.40 | 4.37 | 0.70 | 9.06 | 0.68 | 2.10 | 0.63 | 1.89 | **0.34** | **0.00** | **0.42** | **0.00** |
| CarCircle | 0.68 | 8.84 | 0.49 | 4.48 | 0.44 | 7.73 | 0.71 | 2.19 | **0.49** | **0.73** | 0.40 | 0.11 | **0.50** | **0.00** |
| DroneCircle | 0.95 | 18.56 | -0.27 | 1.29 | 0.24 | 2.19 | 0.55 | 1.29 | 0.54 | 2.36 | **0.48** | **0.00** | 0.36 | 0.07 |
| **BulletGym Average** | 0.71 | 10.63 | 0.32 | 5.28 | 0.55 | 7.64 | 0.63 | 2.09 | 0.56 | 2.44 | **0.39** | **0.10** | **0.39** | **0.08** |

For any state representation $\Theta_1, \Theta_2$, we say $\Theta_1$ is finer than $\Theta_2$, denoted as $\Theta_1 \succeq \Theta_2$, if and only if for any states $s_1, s_2 \in \mathcal{S}$, $\Theta_1(s_1) = \Theta_1(s_2)$ implies $\Theta_2(s_1) = \Theta_2(s_2)$. Givan et al. (2003) established the relationship between bisimulation and the $Q^*$-irrelevance representations ($\Theta_{\text{bisim}} \succeq \Theta_{Q^*}$) for finite-horizon MDPs with respect to the general Bellman operator. In the following theorem, we extend this relationship to infinite-horizon MDPs and incorporate the safety Bellman operator, as described below.

**Theorem 3.3.** *For any MDP, the optimal Q induced by either the general Bellman operator or the safety Bellman operator satisfy $\Theta_{\text{bisim}} \succeq \Theta_{Q^*}$. The optimal policies derived from both bisimulation representation and $Q^*$-irrelevance representation are also optimal in the ground MDP.*

Theorem 3.3 shows that neither of the representations alters the optimal policy, while the bisimulation representation is finer than the Q-based representation, which implies that

$$0 \leq H(s|\Theta_{\text{bisim}}(s)) \leq H(s|\Theta_{Q^*}(s)). \tag{15}$$

Refer to Appendix A for detailed proof. Since our objective is to maximize the conditional entropy $H(s|z_\theta(s))$, our Q-supervised contrastive learning method theoretically surpasses bisimulation in terms of generalization.

## 4. Experiment

### 4.1. Evaluation on DSRL benchmark

We compare the proposed SDQC with several state-of-the-art baseline safe offline RL algorithms on the DSRL benchmark(Liu et al., 2023a), which provides extensive datasets and environment wrappers for safe offline RL performance evaluation. Evaluation results[†] are presented in Table 1. The baseline algorithms include i) BCQ-Lag: A PID-Lagrangian-based method (Stooke et al., 2020) that considers cost threshold based on Batch Constrained Q-learning (BCQ) (Fujimoto et al., 2019), ii) CPQ (Xu et al., 2022a): A constrained Q-updating method that incorporates penalties for OOD actions and unsafe actions, iii) COptiDICE (Lee et al., 2022): A DICE (distribution correction estimation) based Lagrangian method that builds upon OptiDICE (Lee et al., 2021), iv) CDT (Liu et al., 2023b): A future cost inference method based on Decision Transformer (DT) (Chen et al., 2021b), v) TREBI (Lin et al., 2023): A real-time cost budget inference method on the basis of Diffuser (Janner et al., 2022), vi) FISOR (Zheng et al., 2024): An HJ reachability guided method with diffusion policies that firstly considers the hard constraints in safe offline RL problems.

Our ultimate objective is to achieve zero-cost during test, aligning with the framework established by FISOR (Zheng et al., 2024). However, most baseline algorithms struggle to operate effectively under a zero-cost threshold. Consequently, following FISOR (Zheng et al., 2024), we impose a

---

[†]The baseline algorithm evaluation results are sourced from FISOR (Zheng et al., 2024), except for the evaluation of the Point agent on Safety-Gymnasium (marked with *), which is conducted independently as it is not in the source.

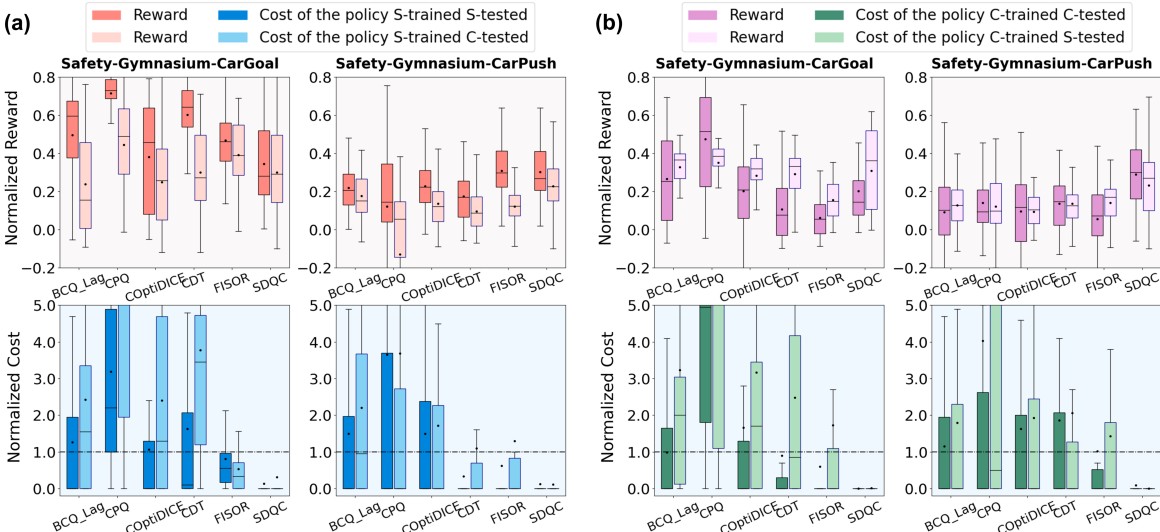

*Figure 3.* The generalization tests on the agent "Car" in Safety-Gymnasium. (a) The agent is trained on the dataset from a simple environment (S-trained), and its performance is evaluated in both the simple environment (S-tested) and the complicated one (C-tested). Conversely, (b) the agent is trained on the dataset from a complicated environment (C-trained), and its performance is assessed in both the original environment (C-tested) and the simple one (S-tested). The evaluation results are obtained from 3 random seeds, with 20 tests on each seed. Outlier data points are omitted for clarity.

stringent cost limit of 10 for the Safety-Gymnasium environment and 5 for Bullet-Safety-Gym. We employ the metrics of *normalized return* and *normalized cost* for evaluation, where a normalized cost below 1 signifies a safe operation.

The former five baseline algorithms exhibit either significant constraint violations or suboptimal returns when subjected to stringent safety requirements, partially due to the fact that they consider only soft constraints. Despite incorporating hard constraints, FISOR still encounters high costs in tasks with high complexity. As discussed in Section 1, this issue can be attributed to estimation errors and OOD problems during the testing phase. In contrast to FISOR, our proposed SDQC conducts safety assessments on the cost-related representation abstracted from the original observations and makes decisions accordingly. The utilization of decoupled representations in SDQC substantially improves the accuracy of state safety assessment and enhances the generalization capability of the policy, thereby providing a higher level of safety assurance. The experimental results clearly demonstrate that SDQC outperforms FISOR in terms of higher rewards and lower costs. Remarkably, SDQC even achieves zero violations in the majority of tasks.

### 4.2. Generalization Tests

To showcase the superior generalization capabilities of our proposed SDQC compared to other safe offline RL algorithms, we perform generalization tests on the "CarGoal" and "CarPush" tasks (in Safety-Gymnasium), as illustrated in Figure 3. In these tasks, the "Car" agent is tasked with reaching the goal point or pushing the box to the goal point

while avoiding hazardous areas and obstacles. The difficulty level varies between tasks, with the simple tasks (CarGoal1, CarPush1) having fewer hazards and obstacles than the challenging tasks (CarGoal2, CarPush2).

It is reasonable to be concerned about the performance of an agent when it is tested in environments that differ from the ones it was trained on, especially if the testing environment is more complex or comprehensive. The experimental results provide evidence that our proposed SDQC algorithm is the only algorithm that ensures no increase in cost under such circumstances. In fact, SDQC achieves almost zero violations in the majority of tests, with only a slight decay in reward performance. In contrast, other algorithms exhibit a sharp increase in cost and/or a significant decrease in reward. Generalization in ensuring safety is crucial in safety-critical scenarios like autonomous driving. It is impractical for the agent to traverse every possible radar observation that may arise in real-world scenarios during training. Our proposed SDQC offers a potent and promising solution for addressing these complex safety-critical scenarios.

### 4.3. Ablation Study

To validate the efficacy of our proposed Q-supervised contrastive learning approach in acquiring meaningful representations and enhancing performance, as discussed in Section 3.2, we conducted ablation studies on the "Safety-Gym-CarGoal2" task (cf. Figure 4). In the absence of contrastive loss during the critic and representation training phase, the agent experiences considerably lower rewards and higher costs compared to the agent trained with contrastive loss.

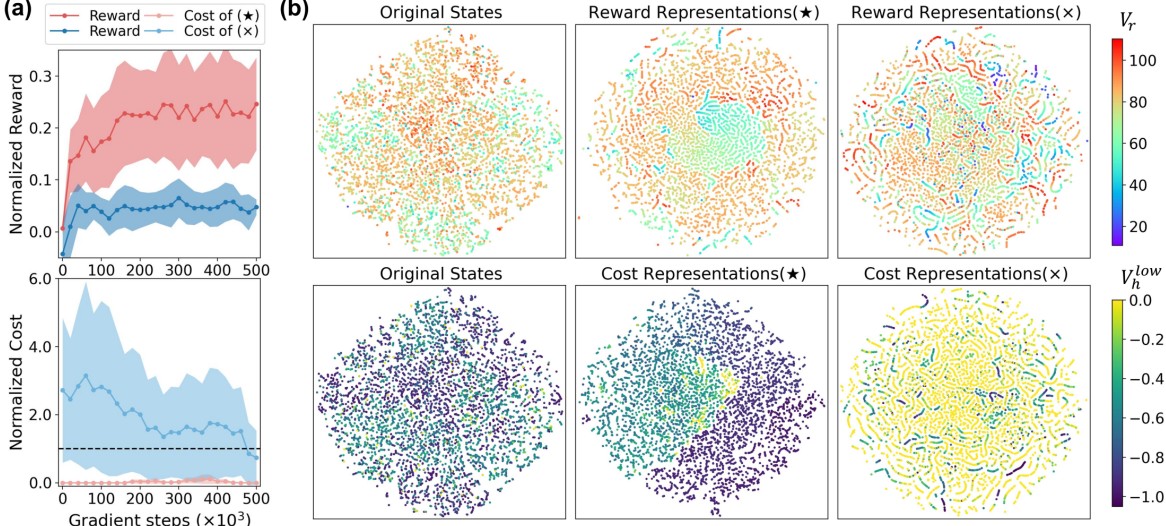

*Figure 4.* Ablation studies on the Q-supervised contrastive loss in CarGoal2. (a) The actor-training-process evaluations of SDQC with (marked by ★) and without (marked by ×) contrastive loss. The curves are averaged over 3 random seeds and smoothed with a window size of 3. (b) t-SNE visualization of the distribution of the original state, the reward-related and the cost-related representations with and without contrastive loss across 10 different safe trajectories, where the policies are from the agent trained with contrastive loss. The original states (first column) are colored according to the critic trained with contrastive loss (i.e., the same as the second column).

The t-SNE visualization results (in Figure 4b) reveal that the Q-supervised contrastive loss effectively clusters representations with similar values. This aligns with our original intention, which is to cluster states with similar Q-values for any actions in the representation space. The clusterings facilitate the learning of the conditional diffusion model (i.e., the actor) by promoting the generation of similar output policies for similar representations. Furthermore, the inclusion of the Q-supervised contrastive loss enables a more reliable evaluation of the states' safety. In the depicted 10 trajectories, despite the cumulative cost being zero, the agent trained without contrastive loss erroneously identifies a majority of the experienced states as unsafe ($V_h^{\text{low}} \geq 0$). Conversely, the agent trained with contrastive loss provides a more accurate assessment, demonstrating the effectiveness of the proposed approach. For more ablation studies on the impact of anchor number ($|\mathcal{I}|$) and neural network structures, please refer to Appendix C.

## 5. Related works

**Safe RL.** In online settings, safe RL problems are generally tackled with two mainstream approaches (Xu et al., 2022b). i) Formulating the problem as a CMDP and solving it from an optimization perspective. Solution techniques include updating the policy constrained in a trust region (Achiam et al., 2017; Liu et al., 2022), reformulating the problem into its Lagrangian dual form (Tessler et al., 2018; Chow et al., 2018; Ma et al., 2021b; Duan et al., 2022), and addressing the constraints by framing an opti-

mistic/pessimistic planning problem (Wachi et al., 2018; Kalagarla et al., 2021). ii) Combining the safe RL problem with the field of safe control. A prevalent method entails representing a safety certificate through a learned safety assessment function, such as the Control Barrier Function (CBF) (Ma et al., 2021a; Luo & Ma, 2021) or Hamilton-Jacobbi (HJ) reachability (Yu et al., 2022; Fisac et al., 2019; Chen et al., 2021a). An agent can switch between optimal and safe policies based on safety assessment results (Chen et al., 2021a; Thananjeyan et al., 2021), thereby theoretically ensuring hard constraints with state-wise zero violations. Recent research endeavors have embraced the integration of safety constraint problems with existing reliable offline reinforcement learning algorithms (Xu et al., 2022a; Lee et al., 2022; Liu et al., 2023b; Lin et al., 2023). However, most existing methods only provide soft constraints without any guarantees of zero violations. FISOR (Zheng et al., 2024) is the first safe offline RL algorithm that tackles hard constraints issues, while the limited offline data still makes it difficult to guarantee safety during tests thoroughly. As a complementary algorithm to FISOR, our SDQC decouples the global observations for safe decision-making, substantially improves the accuracy of state safety assessment, and enhances the generalization capability of the policy, thereby providing a higher level of safety assurance.

**Representation Learning.** Representation learning in RL involves compressing the large observation space into a smaller latent vector that captures relevant aspects of the environment (Watter et al., 2015; Finn et al., 2016; Gelada et al., 2019), often applied in image-based tasks (Kostrikov

et al., 2020; Yarats et al., 2021; Cetin et al., 2022). Contrastive learning has been widely acknowledged as a potent technique for unsupervised representation learning (Liu et al., 2021; Zhu et al., 2022), primarily achieved by augmenting data through introducing noise to the original image (Laskin et al., 2020; Agarwal et al., 2021). In state-based tasks, this approach is not directly applicable as the noise may distort the underlying information. Unlike previous works that conduct contrastive learning among the generated samples, we employ contrastive learning within the dataset itself in a Q-supervised manner. The most relevant works to ours are from Bellemare et al. (2019) and Le Lan et al. (2021), who learn representations via Bellman value functions. To the best of our knowledge, we are pioneers in utilizing representation learning in state-based Safe RL tasks. We are the first to introduce the concept of decoupling states into reward- and cost-related representations specifically for decision-making purposes.

## 6. Conclusion

In this work, we propose the first framework of state decoupling for safe decision-making to tackle the OOD problem of offline safe RL during the testing phase. We propose a Q-supervised contrastive learning method to learn the representations without relying on additional system model estimation such as bisimulation. Theoretical analysis demonstrates that our Q-supervised approach generates coarser representations while preserving the optimal policy, leading to enhanced generalization performance. Experiments on DSRL benchmarks showcase that SDQC surpasses other baseline algorithms, especially for its exceptional ability to achieve almost zero violations in more than half of tasks. Further, SDQC possesses superior generalization ability when confronted with unseen environments.

### Acknowledgments

This work was supported in part by the General Research Fund (GRF) project 14200720 of the Hong Kong University Grants Committee and the National Natural Science Foundation of China (NSFC) Project 62073273. The authors would like to thank the anonymous reviewers for valuable discussion.

### Impact Statement

The impact of our work lies in its potential to develop trustworthy decision-making systems and advance the field of safe offline RL. Focusing on the hard constraint problems, our SDQC achieves almost zero violations in the DSRL benchmark tasks and possesses great generalization capability to unfamiliar global observations. Its superior performance and adaptability significantly enhance the safety and reliability of RL applications in real-world scenarios, such as autonomous driving, robot control, and industrial management, where safe decision-making holds exceptional significance.

We would like to emphasize that our SDQC does not have any negative societal impacts. Our proposed approach is benign, free from potential malicious or unintended uses, and raises no concerns regarding fairness or privacy.

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

# A. Theoretical Interpretations

The first comparison between the bisimulation representation and $Q^*$-irrelevance representation for finite horizon MDPs was given by Givan et al. (2003). The systematic state abstraction theory for MDPs was summarized in Li et al. (2006). The expansion of the theory in Partially Observable MDPs (POMDPs) are introduced recently (Subramanian et al., 2022). It is worth noting that their formulation does not incorporate the safety Bellman operator, and a comprehensive proof for infinite-horizon MDPs is not provided. We now provide a complete proof for Theorem 3.3 as follows.

**Definition A.1.** For any state representation $\Theta_1, \Theta_2$, we say $\Theta_1$ is finer than $\Theta_2$ (or $\Theta_2$ is coarser than $\Theta_1$), denoted as $\Theta_1 \succeq \Theta_2$, if and only if for any states $s_1, s_2 \in \mathcal{S}$, $\Theta_1(s_1) = \Theta_1(s_2)$ implies $\Theta_2(s_1) = \Theta_2(s_2)$.

To clarify, let $z_1$ and $z_2$ represent the representations $\Theta_1(s)$ and $\Theta_2(s)$ for any $s \in \mathcal{S}$, respectively. It is always possible to find a function $f : \mathcal{Z}_1 \to \mathcal{Z}_2$ that is surjective. The equality holds ($\Theta_1 = \Theta_2$) if and only if the surjective function is also injective (i.e., bijective).

*Theorem* 3.1. *For any MDP, the optimal Q-functions induced by either the general Bellman operator or the safety Bellman operator satisfy $\Theta_{\text{bisim}} \succeq \Theta_{Q^*}$. The optimal policies derived from both bisimulation representation and $Q^*$-irrelevance representation are also optimal in the ground MDP, i.e., $\pi^*(\Theta_{\text{bisim}}(s)) = \pi^*(\Theta_{Q^*}(s)) = \pi^*(s)$ for any state $s \in \mathcal{S}$.*

*Proof.* We start by considering a finite-horizon MDP with a maximum timestep $T$. For any timestep $t \in \{1, 2, ..., T\}$, we denote $Q^*_{r,t(T)}$ as the optimal-Q function at timestep $t$. Then, for $\forall s \in \mathcal{S}$ and $\forall a \in \mathcal{A}$, we have:

$$Q^*_{r,t(T)}(s,a) = r(s,a) + \gamma \sum_{s' \in \mathcal{S}} P(s' \mid s_1, a)[\max_{a' \in \mathcal{A}} Q^*_{r,t+1(T)}(s',a')]. \tag{16}$$

For timestep $T + 1$, we define $Q^*_{r,T+1(T)}(s,a) = 0$ for $\forall s \in \mathcal{S}$ and $\forall a \in \mathcal{A}$, which implies that $Q^*_{r,T(T)}(s,a) = r(s,a)$. Now, for any $s_1, s_2 \in \mathcal{S}$ that are bisimilar (i.e., $\Theta_{bisim}(s_1) = \Theta_{bisim}(s_2)$), we have $Q^*_{r,T(T)}(s_1,a) = Q^*_{r,T(T)}(s_2,a)$. In other words, for any $z' \in \mathcal{Z}_{bisim}$, $\max_{a' \in \mathcal{A}} Q^*_{r,T(T)}(s',a')$ is the same for all $s' \in \Theta^{-1}_{bisim}(z')$.

Considering any $s_1, s_2 \in \mathcal{S}$ that are bisimilar, and for any action $a \in \mathcal{A}$, we perform backward induction on timestep $t$ from $T - 1$ to 1 following the proof sketch of Theorem 5 in Givan et al. (2003):

$$
\begin{aligned}
& Q^*_{r,t(T)}(s_1, a) \\
&= r(s_1,a) + \gamma \sum_{s' \in \mathcal{S}} P(s' \mid s_1, a)[\max_{a' \in \mathcal{A}} Q^*_{r,t+1(T)}(s',a')] \\
&\overset{a}{=} r(s_1,a) + \gamma \sum_{s' \in \{\cup_{z' \in \mathcal{Z}} \Theta^{-1}(z')\}} P(s' \mid s_1, a)[\max_{a' \in \mathcal{A}} Q^*_{r,t+1(T)}(s',a')] \\
&\overset{b}{=} r(s_1,a) + \gamma \sum_{z' \in \mathcal{Z}} \sum_{s' \in \Theta^{-1}(z')} P(s' \mid s_1, a)[\max_{a' \in \mathcal{A}} Q^*_{r,t+1(T)}(s',a')] \\
&\overset{c}{=} r(s_2,a) + \gamma \sum_{z' \in \mathcal{Z}} \sum_{s' \in \Theta^{-1}(z')} P(s' \mid s_2, a)[\max_{a' \in \mathcal{A}} Q^*_{r,t+1(T)}(s',a')] \\
&= Q^*_{r,t(T)}(s_2, a).
\end{aligned}
\tag{17}
$$

Equalities (a) and (b) hold due to the surjective relationship between $s$ and $z$. Equality (c) holds due to the definition of bisimulation and the fact that, for $\forall z' \in \mathcal{Z}_{bisim}$, $\max_{a' \in \mathcal{A}} Q^*_{r,t+1}(s',a')$ is the same for all $s' \in \Theta^{-1}_{bisim}(z')$, as established by the induction hypothesis.

We denote $Q^*_r = Q^*_{r,t(\infty)}$ as the optimal Q function for the infinite-horizon MDP. The uniqueness of $Q^*_r$ is guaranteed by the fixed-point property of Bellman operator. For any $s \in \mathcal{S}$, $a \in \mathcal{A}$, and timestep $t \in \{1, 2, ..., T - 1\}$. The optimal Q value gap between $Q^*_r$ and $Q^*_{r,t(T)}$ can be expressed as:

$$
\begin{aligned}
& \left| Q^*_r(s,a) - Q^*_{r,t(T)}(s,a) \right| \\
&= \Big| r(s,a) + \sum_{s' \in \mathcal{S}} P(s'|s,a)[\gamma \max_{a' \in \mathcal{A}} Q^*_r(s',a')] - \\
& \quad r(s,a) - \sum_{s' \in \mathcal{S}} P(s'|s,a)[\gamma \max_{a' \in \mathcal{A}} Q^*_{r,t+1(T)}(s',a')] \Big|
\end{aligned}
$$

$$= \left| \sum_{s' \in \mathcal{S}} P(s'|s,a) [\gamma \max_{a' \in \mathcal{A}} Q_r^*(s,a) - \gamma \max_{a' \in \mathcal{A}} Q_{r,t+1(T)}^*(s',a')] \right|$$

$$\overset{a}{\leq} \sum_{s' \in \mathcal{S}} P(s'|s,a) \left| [\gamma \max_{a' \in \mathcal{A}} Q_r^*(s',a') - \gamma \max_{a' \in \mathcal{A}} Q_{r,t+1(T)}^*(s',a')] \right|$$

$$\overset{b}{\leq} \sum_{s' \in \mathcal{S}} P(s'|s,a) \gamma \max_{a' \in \mathcal{A}} \left| Q_r^*(s',a') - Q_{r,t+1(T)}^*(s',a') \right|$$

$$\overset{c}{\leq} \gamma \max_{a' \in \mathcal{A}, s' \in \mathcal{S}} \left| Q_r^*(s',a') - Q_{r,t+1(T)}^*(s',a') \right|, \tag{18}$$

where inequality (a) holds due to the triangle inequality property, and inequalities (b) and (c) follow from the properties of the maximum function. Assuming that $Q_r^*$ and $Q_{r,T(T)}^*$ are bounded for any $s \in \mathcal{S}$ and $a \in \mathcal{A}$, we conclude that for a discount factor $\gamma \in (0,1)$, the difference $|Q_r^*(s,a) - Q_{r,t(T)}^*(s,a)|$ converges to 0 as $T \to \infty$. Applying backward induction as introduced in Eq. (17), we deduce that, for any $s_1, s_2 \in \mathcal{S}$ that are bisimilar (i.e., $\Theta_{bisim}(s_1) = \Theta_{bisim}(s_2)$), $Q_r^*(s_1, a) = Q_r^*(s_2, a)$. This completes the proof that $\Theta_{bisim} \succeq \Theta_{Q^*}$ for general Bellman operators.

For the safety Bellman operator, we have an analogous definition that for any timestep $t \in \{1, 2, ..., T\}$, $Q_{h,t(T)}^*$ is the optimal-Q function at timestep $t$. Then, for $\forall s \in \mathcal{S}$ and $\forall a \in \mathcal{A}$, we have:

$$Q_{h,t(T)}^*(s,a) = (1-\gamma)h(s) + \gamma \sum_{s' \in \mathcal{S}} P(s' \mid s,a)[\max\{h(s), \min_{a' \in \mathcal{A}} Q_{h,t+1(T)}^*(s',a')\}]. \tag{19}$$

We denote $Q_{h,T+1(T)}^*(s,a) = h(s)$ for all $s \in \mathcal{S}$ and $a \in \mathcal{A}$, which implies that $Q_{h,T(T)}^*(s,a) = h(s)$ if $h(s) > 0$ and $Q_{h,T(T)}^*(s,a) = (1-\gamma)h(s)$ otherwise. Given any state $s \in \mathcal{S}$, for all $z' \in \mathcal{Z}_{bisim}$, $\max\{h(s), \min_{a' \in \mathcal{A}} Q_{h,T(T)}^*(s',a')\}$ is the same for all $s' \in \Theta_{bisim}^{-1}(z')$. For any $s_1, s_2 \in \mathcal{S}$ that are bisimilar, and for any action $a \in \mathcal{A}$, we apply backward induction from timestep $T-1$ to 1 such that:

$$Q_{h,t(T)}^*(s_1, a)$$
$$= (1-\gamma)h(s_1) + \gamma \sum_{s' \in \{\cup_{z' \in \mathcal{Z}} \Theta^{-1}(z')\}} P(s' \mid s_1, a)[\max\{h(s_1), \min_{a' \in \mathcal{A}} Q_{h,t+1(T)}^*(s',a')\}]$$
$$= (1-\gamma)h(s_1) + \gamma \sum_{z' \in \mathcal{Z}} \sum_{s' \in \Theta^{-1}(z')} P(s' \mid s_1, a)[\max\{h(s_1), \min_{a' \in \mathcal{A}} Q_{h,t+1(T)}^*(s',a')\}] \tag{20}$$
$$= (1-\gamma)h(s_2) + \gamma \sum_{z' \in \mathcal{Z}} \sum_{s' \in \Theta^{-1}(z')} P(s' \mid s_2, a)[\max\{h(s_2), \min_{a' \in \mathcal{A}} Q_{h,t+1(T)}^*(s',a')\}]$$
$$= Q_{h,t(T)}^*(s_2, a).$$

Similarly, we denote $Q_h^* = Q_{h,t(\infty)}^*$ as the optimal Q function with safety Bellman operator for the infinite-horizon MDP. The uniqueness of $Q_h^*$ is also guaranteed by the fixed-point property of safety Bellman operator. For any $s \in \mathcal{S}$, $a \in \mathcal{A}$, and timestep $t \in \{1, 2, ..., T-1\}$. The optimal Q value gap between $Q_h^*$ and $Q_{h,t(T)}^*$ can be expressed as:

$$\left| Q_h^*(s,a) - Q_{h,t(T)}^*(s,a) \right|$$
$$= \left| (1-\gamma)h(s) + \sum_{s' \in \mathcal{S}} P(s'|s,a)[\gamma \max\{h(s), \min_{a' \in \mathcal{A}} Q_h^*(s',a')\} - \right.$$
$$\left. (1-\gamma)h(s) - \sum_{s' \in \mathcal{S}} P(s'|s,a)[\gamma \max\{h(s), \min_{a' \in \mathcal{A}} Q_{h,t+1(T)}^*(s',a')\} \right|$$
$$= \left| \sum_{s' \in \mathcal{S}} P(s'|s,a)[\gamma \max\{h(s), \min_{a' \in \mathcal{A}} Q_h^*(s',a')\} - \gamma \max\{h(s), \min_{a' \in \mathcal{A}} Q_{h,t+1(T)}^*(s',a')\}] \right|$$
$$\leq \sum_{s' \in \mathcal{S}} P(s'|s,a) \left| \gamma \max\{h(s), \min_{a' \in \mathcal{A}} Q_h^*(s',a')\} - \gamma \max\{h(s), \min_{a' \in \mathcal{A}} Q_{h,t+1(T)}^*(s',a')\} \right|$$

$$\leq \sum_{s' \in \mathcal{S}} P(s'|s,a)\gamma \max_{a' \in \mathcal{A}} \left| Q_h^*(s',a') - Q_{h,t+1(T)}^*(s',a') \right|$$

$$\leq \gamma \max_{a' \in \mathcal{A}, s' \in \mathcal{S}} \left| Q_h^*(s',a') - Q_{h,t+1(T)}^*(s',a') \right|. \tag{21}$$

A similar conclusion can be drawn that for a discount factor $\gamma \in (0,1)$, the difference $|Q_h^*(s,a) - Q_{h,t(T)}^*(s,a)|$ approaches to 0 as $T \to \infty$, provided that $Q_r^*$ and $Q_{r,T(T)}^*$ are bounded for any $s \in \mathcal{S}$ and $a \in \mathcal{A}$. With backward induction as introduced in Eq. (20), we conclude that $\Theta_{bisim} \succeq \Theta_{Q^*}$ for safety Bellman operators.

**Lemma A.2.** *For any MDP, given the optimal Q-functions induced by either the general Bellman operator or the safety Bellman operator, the optimal policy for $\Theta_{Q^*}$ remains optimal in the ground MDP.*

*Proof.* For any state $s \in \mathcal{S}$, we observe that $Q^*(\Theta_{Q^*}(s),a) = Q^*(s,a)$ for any $a \in \mathcal{A}$, as per Definition 3.2. It is evident that for a given state $s \in \mathcal{S}$:

$$a_r^* = \arg\max_{a \in \mathcal{A}} Q_r^*(\Theta_{Q_r^*}(s),a) = \arg\max_{a \in \mathcal{A}} Q_r^*(s,a), \tag{22}$$

$$a_h^* = \arg\min_{a \in \mathcal{A}} Q_h^*(\Theta_{Q_h^*}(s),a) = \arg\min_{a \in \mathcal{A}} Q_h^*(s,a). \tag{23}$$

Therefore, we conclude that the optimal policy is preserved for $Q^*$-irrelevant representations.

**Lemma A.3.** *For any MDP, given the optimal Q-functions induced by either the general Bellman operator or the safety Bellman operator, and any representation $\Theta_1$ that is finer than $\Theta_{Q^*}$, i.e. $\Theta_1 \succeq \Theta_{Q^*}$, it holds that $Q^*(\Theta_1(s),a) = Q^*(s,a)$ for any $s \in \mathcal{S}$ and $a \in \mathcal{A}$. The optimal policy for $\Theta_1$ is also optimal in the ground MDP.*

*Proof.* We denote the optimal value function for representation $\Theta_1, \Theta_{Q^*}$ and ground state $s$ as $Q_{\Theta_1}^*, Q_{\Theta_{Q^*}}^*$ and $Q_s^*$, respectively. It is evident that $Q_{\Theta_1}^*(\Theta_1(s),a) = Q_{\Theta_{Q^*}}^*(\Theta_{Q^*}(s),a) = Q_s^*(s,a)$ for any $s \in \mathcal{S}, a \in \mathcal{A}$ is one of the solutions for $Q_{\Theta_1}^*$, due to the subjective relationship between $\Theta_1(s)$ and $\Theta_{Q^*}(s)$. To show that the optimal value function for representation $\Theta_1$ is unique, suppose that there exist two optimal value functions $Q_{r1}^*(z,a)$ and $Q_{r2}^*(z,a)$ for any representation $z \in \mathcal{Z}_{\Theta_1}$ and action $a \in \mathcal{A}$. The gap between them can be expressed as

$$\begin{aligned} \Delta_{Q_r^*}(z,a) &= |Q_{r1}^*(z,a) - Q_{r2}^*(z,a)| \\ &= \left| \sum_{z'} P(z'|z,a)\gamma \left( \max_{a'} Q_{r1}^*(z',a') - \max_{a'} Q_{r2}^*(z',a') \right) \right| \\ &\leq \gamma \max_{z',a'} \Delta_{Q_r^*}(z',a'), \end{aligned} \tag{24}$$

where the inequality directly arises from the reasoning outlined in Eq. (18). With the discount factor $\gamma \in (0,1)$, $\Delta_{Q_r^*}(z,a)$ tends to zero for any finite value of $Q_{r1}^*(z,a)$ and $Q_{r2}^*(z,a)$. This implies that the fixed point $Q^*$ for the general Bellman operator is always unique.

For the safety Bellman operators, we also have

$$\begin{aligned} \Delta_{Q_h^*}(z,a) &= |Q_{h1}^*(z,a) - Q_{h2}^*(z,a)| \\ &= \left| \sum_{z'} P(z'|z,a)\gamma \left( \max\{h(z), \min_{a'} Q_{h1}^*(z',a')\} - \max\{h(z), \min_{a'} Q_{h1}^*(z',a')\} \right) \right| \\ &\leq \gamma \max_{z',a'} \Delta_{Q_h^*}(z',a'), \end{aligned} \tag{25}$$

where the inequality is a straightforward result of the proof sketch given in Eq. (21). It can be concluded that $Q^*(\Theta_1(s),a) = Q^*(s,a)$ holds for any $\Theta_1 \succeq \Theta_{Q^*}$, both for the general Bellman operator and the safety Bellman operator. Therefore, the optimal policy for $\Theta_1$ is also optimal in the ground MDP, following the proof sketch provided in Lemma A.2.

Combining Eqs. 17 18 and 20 21, we conclude that $\Theta_{bisim} \succeq \Theta_{Q^*}$ holds for both the general Bellman operator and the safety Bellman operator. Combining Lemma A.2 and Lemma A.3, we conclude that both $\Theta_{bisim}$ and $\Theta_{Q^*}$ preserve optimality for the ground MDP. The proof of Theorem 3.1 is complete.

Theorem 3.1 shows that our $Q^*$-irrelavance representation leads to smaller representation space than bisimulation representation, while preserving the optimal policy. A smaller representation space typically implies higher generalization capabilities and higher sampling efficiency during the policy learning process (Liu et al., 2021).

**Proposition A.4.** *For any MDP, $\Theta_1 \succeq \Theta_2$ indicates $H(s|\Theta_1(s)) \leq H(s|\Theta_2(s))$.*

*Proof.* We denote $z_1$ and $z_2$ as representations of $\Theta_1(s)$ and $\Theta_2(s)$, respectively. We have

$$
\begin{aligned}
H(s|z_1) &= -\sum_{s,z_1} p(s,z_1) \log \frac{p(s,z_1)}{p(z_1)} \\
&= -\sum_{s,z_1} p(s,z_1) \log p(s,z_1) + \sum_s \sum_{z_1} p(z_1)p(s|z_1) \log p(z_1) \\
&\stackrel{a}{=} -\sum_s p(s) \log p(s) + \sum_{z_1} p(z_1) \log p(z_1) \\
&\stackrel{b}{=} -\sum_s p(s) \log p(s) + \sum_{z_2} p(z_2) \sum_{z_1} p(z_1|z_2) \log p(z_1) \\
&\stackrel{c}{\leq} -\sum_s p(s) \log p(s) + \sum_{z_2} p(z_2) \log \sum_{z_1} p(z_1|z_2)p(z_1) \\
&\stackrel{d}{\leq} -\sum_s p(s) \log p(s) + \sum_{z_2} p(z_2) \log \sum_{z_1} \mathbb{I}_{\{p(z_1|z_2)\neq 0\}} p(z_1) \\
&\stackrel{e}{=} -\sum_s p(s) \log p(s) + \sum_{z_2} p(z_2) \log p(z_2) \\
&= H(s|z_2),
\end{aligned}
\tag{26}
$$

where $\mathbb{I}_{\{p(z_1|z_2)\neq 0\}} = 1$ if $p(z_1|z_2) \neq 0$, and $\mathbb{I}_{\{p(z_1|z_2)\neq 0\}} = 0$ otherwise. Equality $(a)$ holds as $z_1$ is a function of $s$. Note that $\sum_{z_1 \in \mathcal{Z}_1} p(z_1|z_2) = 1$ for $\forall z_2 \in \mathcal{Z}_2$ in equality $(b)$. Inequality $(c)$ is a consequence of Jensen's inequality. Inequality $(d)$ holds since for $\forall z_2 \in \mathcal{Z}_2$, the conditional probability $p(z_1|z_2)$ does not exceed 1 for all $z_1 \in \{z \in \mathcal{Z}_1 | p(z|z_2) \neq 0\}$. Equality $(e)$ holds due to the surjective relationship between $z_1$ and $z_2$.

Based on Theorem 3.1 and Proposition A.4, we conclude that:

$$
0 \leq H(s|\Theta_{\text{bisim}}(s)) \leq H(s|\Theta_{Q^*}(s)).
\tag{27}
$$

Given our primary objective of maximizing the conditional entropy $H(s|z_\theta(s))$, the proposed Q-supervised contrastive learning method theoretically exhibits superior generalization capabilities compared to bisimulation.

# B. Methodology Clarifications

## B.1. Implicit Q-Learning

Implicit Q-Learning (IQL) is the pioneering in-sample offline RL algorithm proposed by Kostrikov et al. (2021). It decouples the estimation of optimal Q-values from policy optimization, enabling implicit policy learning through the value function. Unlike standard Q-learning, which explicitly derives a policy by maximizing Q-values, IQL avoids direct maximization, reducing susceptibility to instability issues such as overestimation or divergence. The core technique in IQL is upper expectile regression. Given a random variable $X$ with an unknown distribution, the $\tau \in (0, 1)$ expectile can be estimated by solving:

$$
\arg\min_{m_\tau} \mathbb{E}_{x \sim X}[L^\tau(x - m_\tau)], \text{ where } L^\tau(u) = |\tau - \mathbb{I}(u < 0)|u^2
\tag{28}
$$

Specifically, as $\tau \to 1$, the solution to Eq. (28) approximates the upper bound of the random variable $X$. Extending this to conditional distributions, the optimal value function can be approximated by minimizing:

$$
\mathcal{L}_V = \mathbb{E}_{(s,a) \sim D_\beta} \left[ L^\tau(Q(s,a) - V(s)) \right],
\tag{29}
$$

and the optimal Q function can be updated accordingly with the TD loss

$$
\mathcal{L}_Q = \mathbb{E}_{(s,a,s',r) \sim D_\beta} \left[ (r + \gamma V(s') - Q(s,a))^2 \right].
\tag{30}
$$

This process yields in-support optimal value and Q functions without training optimal policies. In this paper, we extend IQL with safety Bellman Operator and estimate both upper bound and lower bound of the value functions for safety assessments.

## B.2. Diffusion Behavior Cloner

The diffusion model was initially introduced as an iterative denoising framework for image generation in the domain of *computer vision* (Sohl-Dickstein et al., 2015; Ho et al., 2020). More recently, it has been adapted for decision-making in state-based tasks, due to its superior performance in capturing action distributions within a dataset. As introduced in Section 3.2, SDQC requires a behavior cloner to capture and reproduce in-support actions for each state within the offline datasets. Following the approach of Lu et al. (2022a;b), we employ score-based diffusion and the DPM-Solver. The training loss for the behavior cloner $\pi_{behav}$ is expressed as follows:

$$\mathcal{L}_{\pi_{behav}} = \mathbb{E}_{t \sim U(1,T), \zeta \sim \mathcal{N}(0,I), (s,a) \sim D_\beta}[\|\zeta - \zeta_{\psi_{behav}}(a_t, s, t)\|]. \tag{31}$$

After training converges, we use second-order DPM-Solver (Lu et al., 2022a) to form $\pi_{behav}$ and sample $|\mathcal{A}_\beta^s|$ actions for each state in offline datasets $\mathcal{D}_\beta$. These actions will be utilized in the subsequent joint optimization of Q-functions and representations.

## B.3. Diffusion Policy

As described in Eqs. 13 and 14, we train three distinct diffusion policies using weighted regression (Zheng et al., 2024). In line with most existing diffusion-based policies (Wang et al., 2022; Garg et al., 2023; Lu et al., 2023), our three policies can be formulated as follows:

$$\begin{cases} \pi_r(a|z_{\theta_r}(s)) = p_{\psi_r}(a_{0:T}|z_{\theta_r}(s)) = \mathcal{N}(a_T; \mathbf{0}, \mathbf{I}) \prod_{t=1}^T p_{\psi_r}(a_{t-1}|a_t, z_{\theta_r}(s)) \\ \pi_h(a|z_{\theta_h}(s)) = p_{\psi_h}(a_{0:T}|z_{\theta_h}(s)) = \mathcal{N}(a_T; \mathbf{0}, \mathbf{I}) \prod_{t=1}^T p_{\psi_h}(a_{t-1}|a_t, z_{\theta_h}(s)) \\ \pi_{to}(a|z_{\theta_r}(s), z_{\theta_h}(s)) = p_{\psi_{to}}(a_{0:T}|z_{\theta_r}(s), z_{\theta_h}(s)) \\ \qquad = \mathcal{N}(a_T; \mathbf{0}, \mathbf{I}) \prod_{t=1}^T p_{\psi_{to}}(a_{t-1}|a_t, z_{\theta_r}(s), z_{\theta_h}(s)), \end{cases} \tag{32}$$

where the reverse transitions are modeled as Gaussian process:

$$\begin{cases} p_{\psi_r}(a_{t-1}|a_t, z_{\theta_r}(s)) = \mathcal{N}(a_{t-1}; \mu_{\psi_r}(a_t, z_{\theta_r}(s), t), \Sigma(t)) \\ p_{\psi_h}(a_{t-1}|a_t, z_{\theta_h}(s)) = \mathcal{N}(a_{t-1}; \mu_{\psi_h}(a_t, z_{\theta_h}(s), t), \Sigma(t)) \\ p_{\psi_{to}}(a_{t-1}|a_t, z_{\theta_r}(s), z_{\theta_h}(s)) = \mathcal{N}(a_{t-1}; \mu_{\psi_{to}}(a_t, z_{\theta_r}(s), z_{\theta_h}(s), t), \Sigma(t)). \end{cases} \tag{33}$$

Given a variance schedule defined by $\beta_t = 1 - \alpha_t$, we proceed to define:

$$\bar{\alpha}_t = \prod_{i=1}^t \alpha_i, \quad \tilde{\beta}_t = \frac{1 - \bar{\alpha}_{t-1}}{1 - \bar{\alpha}_t} \beta_t. \tag{34}$$

The mean of the Gaussian process is then given by:

$$\begin{cases} \mu_{\psi_r}(a_t, z_{\theta_r}(s), t) = \frac{1}{\sqrt{\alpha_t}} (a_t - \frac{\beta_t}{\sqrt{1-\bar{\alpha}_t}} \zeta_{\theta_r}(a_t, z_{\theta_r}(s), t)) \\ \mu_{\psi_h}(a_t, z_{\theta_h}(s), t) = \frac{1}{\sqrt{\alpha_t}} (a_t - \frac{\beta_t}{\sqrt{1-\bar{\alpha}_t}} \zeta_{\theta_h}(a_t, z_{\theta_h}(s), t)) \\ \mu_{\psi_{to}}(a_t, z_{\theta_r}(s), z_{\theta_h}(s), t) = \frac{1}{\sqrt{\alpha_t}} (a_t - \frac{\beta_t}{\sqrt{1-\bar{\alpha}_t}} \zeta_{\theta_{to}}(a_t, z_{\theta_r}(s), z_{\theta_h}(s), t)), \end{cases} \tag{35}$$

and the covariance matrix is expressed as $\Sigma(t) = \tilde{\beta}_t \mathbf{I}$. During the testing phase, actions can be sampled from the reverse diffusion chain for each diffusion policy. To account for safety considerations, we sample multiple actions and select the one with the lowest $Q_h$ value as the final action to be executed (cf. Appendix E.2 for details).

# C. Implementation Details

## C.1. SDQC Summarization

To provide an intuitive understanding the mechanism of SDQC, we present a brief summary of its training and deployment scheme in this subsection.

---

**Algorithm 1** SDQC Training

---

**Phase 1: Behavior cloner training and behavior actions generation**

**Require:** Initial network $\zeta_{\psi_{behav}}$, datasets $\mathcal{D}_\beta = \{s, a, s', r, h\}^N$

1: **for** each iteration $\{s, a\}^{M_1} \sim \mathcal{D}_\beta$ **do**
2:      Update $\zeta_{\psi_{behav}}$ using Eq. (31)
3: **end for**
4: **for** each state $s \sim \mathcal{D}_\beta$ **do**
5:      Given $s$, generate multiple behavior actions $\mathcal{A}_\beta^s$ with $\zeta_{\psi_{behav}}$ using DPM-Solver (Lu et al., 2022a)
6: **end for**
7: **return** Updated datasets $\mathcal{D}_\beta = \{s, \mathcal{A}_\beta^s, a, s', r, h\}^N$

**Phase 2: Joint optimization for value functions and representations**

**Require:** Initial reward-related network $V_r, Q_r, z_{\theta_r}$, cost-related network $V_h^{\mathrm{up}}, V_h^{\mathrm{low}}, Q_h, z_{\theta_h}$, and datasets $\mathcal{D}_\beta = \{s, \mathcal{A}_\beta^s, a, s', r, h\}^N$

8: **for** each iteration $\{s, \mathcal{A}_\beta^s, a, s', r\}^{M_2} \sim \mathcal{D}_\beta$ **do**
9:      Update $V_r, Q_r, z_{\theta_r}$ jointly using Eq. (8)
10: **end for**
11: **for** each iteration $\{s, \mathcal{A}_\beta^s, a, s', h\}^{M_2} \sim \mathcal{D}_\beta$ **do**
12:      Update $V_h^{\mathrm{up}}, V_h^{\mathrm{low}}, Q_h, z_{\theta_h}$ jointly using Eq. (12)
13: **end for**
14: **return** Reward/Cost-related value function and representation networks $V_r, Q_r, z_{\theta_r}, V_h^{\mathrm{up}}, V_h^{\mathrm{low}}, Q_h, z_{\theta_h}$

**Phase 3: Three policies extraction**

**Require:** Initial policy network $\zeta_{\psi_r}, \zeta_{\psi_h}, \zeta_{\psi_{to}}$, fixed pre-trained network $V_r, Q_r, z_{\theta_r}, V_h^{\mathrm{low}}, Q_h, z_{\theta_h}$, and datasets $\mathcal{D}_\beta = \{s, a\}^N$

15: **for** each iteration $\{s, a\}^{M_3} \sim \mathcal{D}_\beta$ **do**
16:      Calculate regression weight for three policies with $V_r, Q_r, z_{\theta_r}, V_h^{\mathrm{low}}, Q_h, z_{\theta_h}$ using Eq. (14)
17:      Update $\zeta_{\psi_r}, \zeta_{\psi_h}, \zeta_{\psi_{to}}$ using Eq. (13)
18: **end for**
19: **return** Three distinct policies $\pi_r, \pi_h, \pi_{to}$

---

As introduced in Section 3, SDQC requires a three-phase training process (see Algorithm 1). The primary objective of the first stage (lines 1-7) is to generate a set of behavior actions $\mathcal{A}_\beta^s$ for each state $s \in \mathcal{D}_\beta$. The set is then utilized in the second stage to measure the similarity between states, represented by $\tilde{d}(s, \tilde{s}) \approx \sup_{a \in \mathcal{A}_\beta^s} |Q^*(z_\theta(s), a) - Q^*(z_\theta(\tilde{s}), a)|$. The second phase (lines 8-14), known as the joint optimization of value functions and representations, employs expectile regression (Kostrikov et al., 2021) to learn in-support optimal value functions while simultaneously using contrastive learning to cluster similar states (measured by $d(s, \tilde{s})$) within the representation space. The final phase (lines 15-19) introduces three distinct policies, which are trained using weighted regression diffusion models as introduced by Zheng et al. (2024). It is important to note that these policies are conditioned on the representation space rather than the ground-truth state space. During deployment, the selection of the specific policy to be adopted is guided by the evaluation of states using the cost-related value function. For further details, please refer to the subsequent paragraph.

Upon completion of training, SDQC is deployed with three distinct policies $\pi_r, \pi_h, \pi_{to}$ derived from the third training phase, along with two representation networks $z_{\theta_h}, z_{\theta_r}$ and three cost-related value functions $V_h^{\mathrm{up}}, V_h^{\mathrm{low}}, Q_h$ from the second training phase (cf. Algorithm 2). Given any state $s$, we initially conduct safety assessments based on cost-related representations. The condition $V_h^{\mathrm{up}}(z_{\theta_h}(s)) > 0$ (line 2) suggests the existence of in-support actions that might lead to unsafe outcomes, necessitating a joint consideration of safety and reward. A positive $V_h^{\mathrm{down}}(z_{\theta_h}(s))$ (line 3) indicates that no action can ensure safety in future trajectories; the agent's primary objective is therefore to exit the unsafe region by deploying policy $\pi_h$, regardless of the reward considerations. Conversely, $V_h^{\mathrm{low}} \leq 0 < V_h^{\mathrm{up}}$ (line 5) reflects a borderline safe condition, requiring the agent to consider both reward and cost, thereby deploying policy $\pi_{to}$. On the other hand, $V_h^{\mathrm{up}}(z_{\theta_h}(s)) \leq 0$ (line 8) confirms absolute safety, obviating the need for the agent to consider cost-related information. An illustration diagram is presented in the right subplot of Figure 2. Note that single action sampled by diffusion model is not trustworthy enough, thereby sampling batch actions and conduct the one with the lowest $Q_h(z_{\theta_h}(s), a)$ leads to safer outcomes (line 11). Please refer to Appendix E.2 for further details.

---

**Algorithm 2** SDQC Deployment

---

**Require:** Three policies $\pi_r, \pi_h, \pi_{to}$, representation network $z_{\theta_h}, z_{\theta_r}$, and cost-related value-functions $V_h^{\text{up}}, V_h^{\text{low}}, Q_h$
  1: **Given** any state $s$
  2: **if** $V_h^{\text{up}}(z_{\theta_h}(s)) > 0$ **then**
       (There exist in-support actions that may lead to unsafe outcomes)
  3:    **if** $V_h^{\text{down}}(z_{\theta_h}(s)) > 0$ **then**
  4:        sample actions: $\{a_i\}^{\text{cand.}} \sim \pi_h(\cdot | z_{\theta_h}(s))$
  5:    **else**
  6:        sample actions: $\{a_i\}^{\text{cand.}} \sim \pi_{to}(\cdot | z_{\theta_h}(s), z_{\theta_r}(s))$
  7:    **end if**
  8: **else**
       (The state is absolute safe)
  9:    sample actions: $\{a_i\}^{\text{cand.}} \sim \pi_r(\cdot | z_{\theta_r}(s))$
10: **end if**
11: **return** Final action $\arg\min_{a \in \{a_i\}^{\text{cand.}}} Q_h(z_{\theta_h}(s), a)$

---

### C.2. SDQC Network Structure

As described in Section 3.3, the representations in our proposed SDQC framework are trained concurrently with the optimal Q value learning process. The neural network structure illustrated in Figure 5 is utilized for training both the reward- and cost-related representations as well as the value functions. The global observation $s$ is encoded into the representation $z$, and the value functions (both $V$ and $Q$) are computed based on this representation with separate multiple-layer-perceptron (MLP) neural networks.

In certain safe RL benchmark problems, it is observed that the majority of dimensions in the global observation share similar physical meanings. For instance, in the Safety-Gymnasium domain, a significant number of dimensions in the global observations correspond to lidar measurements, which provide information about the distances between the agent and the destination or obstacles in specific directions. This reminds us of the self-attention mechanism (Vaswani et al., 2017), which is known for its superior ability to capture relationships among input information that share similar representations in comparison to traditional MLP architectures.

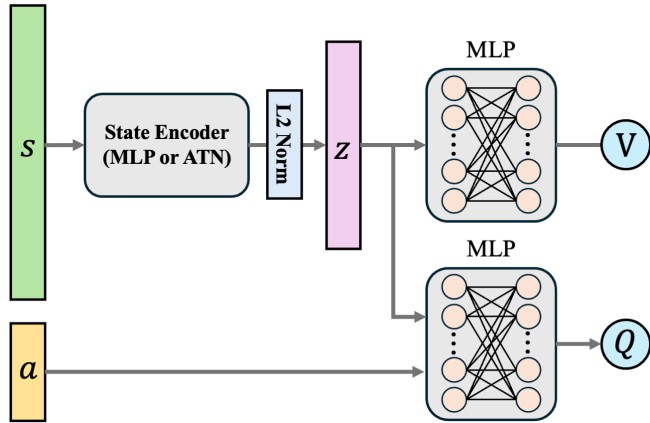

*Figure 5.* Neural network structure for training value functions and representations of SDQC.

Nevertheless, attention mechanisms typically rely on vector multiplication to compute attention weights, which presents a challenge when dealing with global observations where each dimension contains scalar information. Towards this end, we propose to transform each scalar observation dimension into a vector representation using a fixed Gaussian Fourier Encoder (Ho et al., 2020). Subsequently, attention is applied to the encoded vector representations. The output of the attention module is then flattened and passed through an MLP to obtain the final representation. Please refer to Figure 6 for a detailed illustration of the network structure.

To demonstrate the efficacy of our attention-based state encoder (ATN) in effectively capturing information from global observations and identifying the relevance of specific dimensions to reward/cost, we present the attention patterns (i.e., $\text{softmax}(Q \cdot K^T / \sqrt{d_k})$) of the reward- and cost-related state encoders in the task "PointGoal2" (cf. Figure 7). In "PointGoal2", the global observations consist of 60 dimensions. Among them, the first 12 dimensions represent the self-status of the agent, the subsequent 16 dimensions contain reward-related information, and the last 32 dimensions contain cost-related information. Ideally, a reward-related attention pattern should assign higher attention weights to the first 28 dimensions while ignoring the last 32 dimensions. On the other hand, a cost-related attention pattern should focus on the first 12 and last 32 dimensions while disregarding the middle 16 dimensions. The observed attention patterns during our experiments align with the relevance of specific dimensions to reward and cost. For the ablation study on the state encoder network structure, please refer to Appendix D.

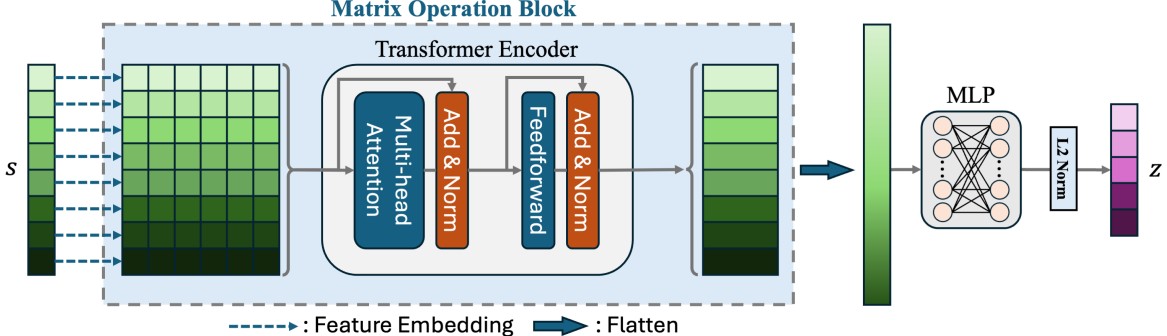

*Figure 6.* Neural network structure of the attention-based state encoder.

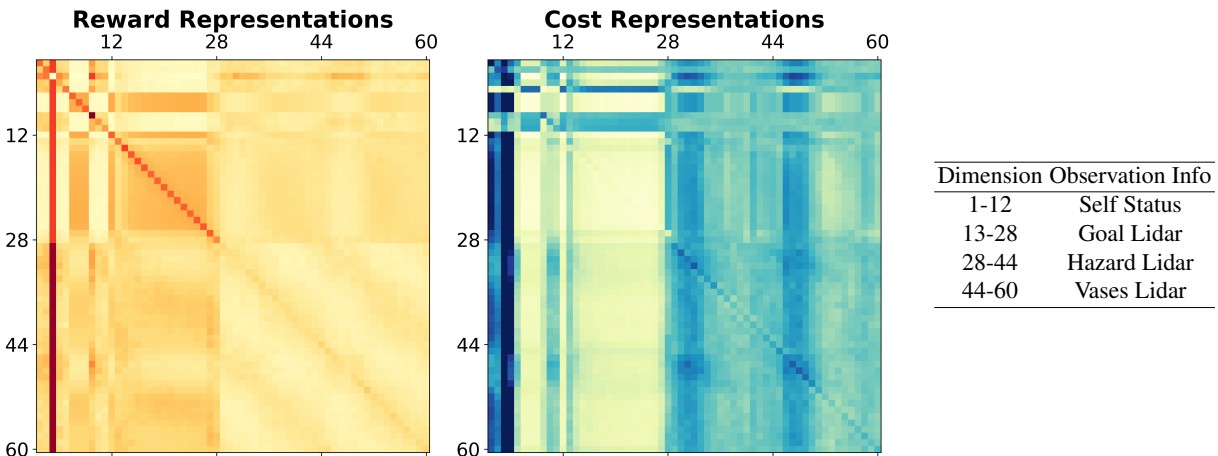

| Dimension | Observation Info |
|-----------|------------------|
| 1-12 | Self Status |
| 13-28 | Goal Lidar |
| 28-44 | Hazard Lidar |
| 44-60 | Vases Lidar |

*Figure 7.* Attention pattern of the reward- and cost-related state encoder, and the meaning of each observation dimension in the task "PointGoal2" from Safety-Gymnasium. Darker colors represent higher values. The "Goal Lidar" dimensions observe the position of the destination, indicating reward information. The "Hazard Lidar" and "Vases Lidar" dimensions observe the position of obstacles, indicating cost information. The pattern is averaged over 3000 observations randomly chosen from DSRL datasets.

### C.3. SDQC Hyperparameters

As discussed in Section 3.2, to calculate the soft similarity measure for contrastive learning, our SDQC framework requires pre-training a generative model to capture the behavior policy of the offline datasets $\pi_\beta$. For this purpose, we employ diffusion probabilistic models (DPM) (Ho et al., 2020; Song et al., 2020) and utilize the DPM-Solver, a fast high-fidelity ODE solver proposed by Lu et al. (2022a;b), to generate the behavior actions of each state $s$ in the offline datasets, denoted as $\mathcal{A}_\beta^s$. We utilize the default network configurations outlined in (Lu et al., 2022a; 2023). Specific hyperparameter settings can be found in Table 2.

SDQC can simultaneously train the reward and cost value functions and their respective representations using Eqs. 8 and 12. The network structures are described in detail in Appendix C.2, and generic hyperparameters can be found in Table 3. Regarding the updating of the safety Bellman operator, we follow the settings in FISOR (Zheng et al., 2024). The constraint violation function is defined as $h(s) = -1$ when the cost function satisfies $c(s) = 0$, and $h(s) = 25$ when $c(s) > 0$.

*Table 2.* Hyperparameters of the DPM-solver for generating behavior actions.

| Hyperparameters | Value |
|-----------------|-------|
| Learning rate | 3e-4 |
| Batch size | 4096 |
| Training steps | 5e5 |
| Diffusion timesteps | 15 |
| Generated action numbers for each state $\|\mathcal{A}_\beta^s\|$ | 8 |

Considering the significant variation in physical meanings among the observation dimensions of different tasks, we employ different state encoder structures accordingly. For tasks that have observation dimensions with diverse physical meanings,

*Table 3.* Generic hyperparameters of SDQC in value functions and representations training phase.

| Module | | Hyper-parameters | Value |
|---|---|---|---|
| General | | Optimizer | Adam |
| | | Learning rate | 3e-4 |
| | | Batch size | 512 |
| | | Training steps | 5e5 |
| | | Soft measure temperature factor $\eta$ | 1.0 |
| | | Contrastive temperature factor $\nu$ | 0.1 |
| | | Contrastive term coef $\delta$ | 1.0 |
| Critic | | Number of hidden layers (Q & V) | 2 |
| | | Number of neurons in hidden layer (Q & V) | 256 |
| | | Activation function (Q & V) | Mish |
| | | Expectile $\tau$ | 0.9 |
| | | Discount factor $\gamma$ | 0.99 |
| | | Target critic soft update | 0.005 |
| State Encoder | MLP | Number of hidden layers | 2 |
| | | Number of neurons in hidden layer | 256 |
| | | Activation function | Mish |
| | ATN | Number of head | 2 |
| | | Embed dimension for each head | 64 |
| | | Dropout rate | 0.1 |

we utilize the MLP structure. Conversely, for tasks where most observation dimensions have consistent physical meanings, we utilize the attention-based state encoder (ATN).

It is observed that the performance of SDQC with an ATN-based state encoder improves when trained with a larger contrastive loss coefficient ($\delta$ in Eqs. 8 and 12) and a higher number of anchor points ($|\mathcal{I}|$ in Eq. (5)). On the other hand, the SDQC performs better with smaller values of $\delta$ and fewer anchor points if the MLP based encoder is used. Besides, the global observation dimensions vary across different tasks. For the ATN-based state encoder, we select the encoded state dimension (i.e., the dimensionality of $z$) to be approximately half of the global observations. On the other hand, for MLP, we choose the encoded state dimension to be roughly twice the size of the global observations. Optimal hyperparameters achieving the best performance on different tasks are presented in Table 4.

*Table 4.* Hyperparameters of SDQC for different tasks. "All" denotes all different tasks for the same agent, while "Vel" refers to the velocity task.

| Domain | Agent | Task | State Encoder | Encoded State Dim | Contrastive Loss Coef | Anchor Number |
|---|---|---|---|---|---|---|
| Safety Gymnasium | Point | All | ATN | 32 | 1.0 | 8 |
| | Car | All | | | | |
| | HalfCheetah | Vel | MLP | 32 | 0.5 | 4 |
| | Ant | Vel | | | | |
| | Swimmer | Vel | | | | |
| Bullet Safety | Ball | All | MLP | 16 | 0.1 | 4 |
| | Car | All | | 32 | | |
| | Drone | All | | 64 | | |
| | Ant | All | | 64 | | |

For the final training phase, which involves policy extraction using weighted regressed diffusion models as described in Eq. (13), we follow the network structure design and generic diffusion parameter selection described by Zheng et al. (Zheng et al., 2024). We train three separate policies ($\pi_r$ $\pi_h$ and $\pi_{to}$) with a learning rate of 0.0003, a batch size of 1024, and the total number of training steps is set to 500,000. The temperature parameters that control the strength of behavior

regularization (in Eq. (14)) are chosen as $\iota_r = \iota_{to} = 3.0$ and $\iota_h = 5.0$.

## D. Additional Ablation Studies

**Ablation studies on network structure.** To demonstrate the effectiveness of our proposed attention-based state encoder (cf. Appendix C.2), we conduct ablation studies on the neural network architecture, as depicted in Figure 8. By substituting the attention-based state encoder with an MLP-based counterpart, we observe a deterioration in the performance of the SDQC, in terms of diminished rewards and increased costs. The t-SNE visualization results depicted in Figure 8b demonstrate that while the MLP-based state encoder does cluster representations with similar values in the high-dimensional space, the clustering effect is not as robust as that achieved by the attention-based approach. Consequently, this leads to an overestimation of the cost value, resulting in inaccurate assessments of the safety condition.

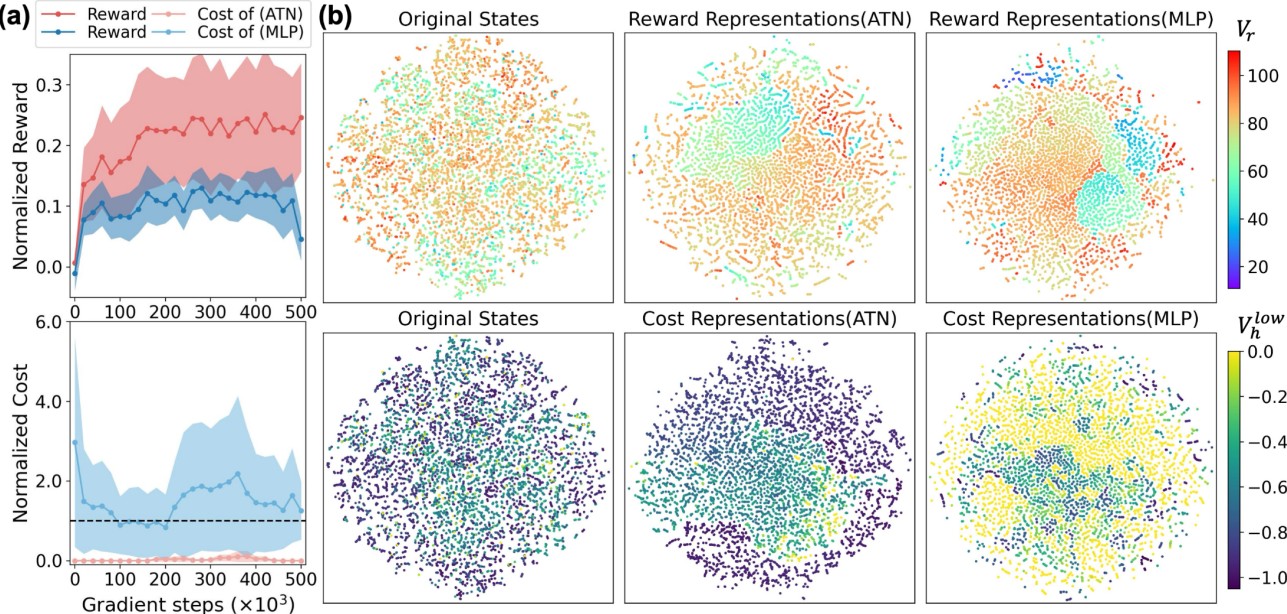

*Figure 8.* Ablation studies on the network structure in CarGoal2. (a) The actor-training-process evaluations of SDQC with attention-based (ATN) and MLP-based (MLP) state encoder. (b) t-SNE visualization of the distribution of the original state, the reward- and cost-related representations with ATN/MLP state encoders across 10 different safe trajectories.

*Table 5.* Ablation studies on the choice of anchor number.

| Anchor Number | CarGoal2 | | | CarPush2 | | |
|---|---|---|---|---|---|---|
| | Reward | Cost | Runtime (s/epoch) | Reward | Cost | Runtime (s/epoch) |
| 16 | 0.22 | 0.00 | 31.4 | 0.28 | 0.06 | 36.2 |
| 8 | 0.23 | 0.00 | 28.8 | 0.31 | 0.04 | 32.7 |
| 4 | 0.20 | 0.05 | 26.0 | 0.25 | 0.06 | 29.9 |
| 1 | 0.15 | 0.13 | 23.6 | 0.10 | 0.18 | 27.6 |
| 0 | 0.05 | 0.86 | 19.4 | 0.21 | 2.15 | 23.7 |

**Ablation studies on anchor number choice.** An essential hyperparameter in our proposed Q-supervised contrastive learning method is the anchor number, $|\mathcal{I}|$ in Eq. (5). This parameter determines the number of representation pairs to be clustered in the high-dimensional space during each gradient step. The ablation study results are summarized in Table 5. Our experimentation reveals that an anchor number can result in subpar clustering outcomes, consequently impairing the performance of SDQC. Conversely, overly large anchor numbers lead to an excessive influence of the contrastive loss term

in the overall loss function, increasing the computational costs. To strike a balance and attain optimal performance, we let $|\mathcal{I}| = 8$ for our attention-based state encoders.

**Ablation studies on contrastive-related hyperparameters.** In our SDQC framework, one of the critical components is the contrastive representation loss, as described in Equations 8 to 12. This involves selecting appropriate values for the term coefficient $\delta$ and the exponential temperature $\nu$. As shown in Tables 3 and 4, we vary $\delta$ across different domains but maintain a consistent $\nu = 0.1$ across all environments. The effects of these parameter choices are detailed in Table 6. With respect to the temperature $\nu$, employing a very small value (0.01) tends to destabilize the training process, ultimately resulting in collapse. Conversely, using a larger value (1.0) produces poorly clustered representations, leading to a marked degradation in performance. Regarding the term coefficient $\delta$, a smaller value results in a slight performance decline. However, a larger coefficient excessively prioritizes the contrastive loss, destabilizing the training of the value function and significantly degrading performance. While fine-tuning these hyperparameters for specific environments and tasks could potentially yield better experimental results on the benchmark, we choose not to do so.

*Table 6.* Ablation studies on contrastive-related hyperparameters.

| Env | Contrast Coef. ($\delta$) | Contrast Temp. ($\nu$) | Reward | Cost | Env | Contrast Coef. | Contrast Temp. | Reward | Cost |
|---|---|---|---|---|---|---|---|---|---|
| | 1 | 0.01 | NaN | NaN | | 1 | 0.01 | NaN | NaN |
| | 1 | 0.1 | 0.31 | 0.04 | | 1 | 0.1 | 0.29 | 0.09 |
| PointGoal2 | 1 | 1 | 0.22 | 0.16 | CarPush2 | 1 | 1 | 0.20 | 0.48 |
| | 0.1 | 0.1 | 0.31 | 0.17 | | 0.1 | 0.1 | 0.30 | 0.10 |
| | 10 | 0.1 | 0.23 | 0.48 | | 10 | 0.1 | 0.22 | 0.53 |

**Ablation studies on the deployment of three distinct policies** As introduced in Section 3.3 and Appendix C.1, SDQC coordinates three distinct policies, reward policy $\pi_r$, trade-off policy $\pi_{to}$, and cost policy $\pi_h$, to ensure excellent safety performance. To verify the necessity of each policy, we conduct ablation studies examining their individual deployments, with results presented in Table 7. Notably, a naive reward policy $\pi_r$ focuses solely on maximizing rewards while ignoring costs, a naive cost policy $\pi_h$ prioritizes minimizing costs but disregards rewards, and a naive trade-off policy $\pi_{to}$ takes both into account but fails to excel in either maximizing rewards or minimizing costs. The best performance consistently results from the collaboration of all three policies. When the trade-off policy $\pi_{to}$ is omitted (combining $\pi_r$ and $\pi_h$), the agent incurs higher costs as it cannot respond promptly to borderline dangers. Combining the trade-off policy $\pi_{to}$ and cost policy $\pi_h$ does not increase costs, but results in a decline in reward accumulation. While combining the reward policy $\pi_r$ and trade-off policy $\pi_{to}$ achieves comparable performance to using all three policies, it results in slightly higher costs due to the agent's reduced ability to quickly escape dangerous situations.

*Table 7.* Ablation studies on the deployment of three distinct policies.

| Policy Num. | One | | | | | | Two | | | | | | Three | |
|---|---|---|---|---|---|---|---|---|---|---|---|---|---|---|
| Env Name | Naïve $\pi_r$ | | Naïve $\pi_{to}$ | | Naïve $\pi_h$ | | $\pi_r$ and $\pi_{to}$ | | $\pi_r$ and $\pi_h$ | | $\pi_{to}$ and $\pi_h$ | | $\pi_r$, $\pi_{to}$ and $\pi_h$ | |
| | reward | cost | reward | cost | reward | cost | reward | cost | reward | cost | reward | cost | reward | cost |
| PointGoal1 | 0.69 | 4.92 | 0.27 | 0.69 | 0.01 | 0.00 | 0.32 | 0.42 | 0.34 | 0.51 | 0.22 | 0.18 | 0.35 | 0.36 |
| PointGoal2 | 0.75 | 13.28 | 0.28 | 0.18 | -0.11 | 0.00 | 0.23 | 0.24 | 0.20 | 0.14 | 0.23 | 0.12 | 0.29 | 0.09 |
| CarPush1 | 0.38 | 1.32 | 0.26 | 0.00 | 0.05 | 0.00 | 0.27 | 0.00 | 0.27 | 0.12 | 0.21 | 0.00 | 0.30 | 0.00 |
| CarPush2 | 0.42 | 4.34 | 0.27 | 0.01 | 0.02 | 0.00 | 0.31 | 0.01 | 0.28 | 0.16 | 0.18 | 0.03 | 0.31 | 0.04 |

# E. Experimental Details

## E.1. Task Description

**Safety-Gymnasium (Ray et al., 2019).** A collection of environments based on the Mujoco physics simulator. In the obstacle avoidance series environments, there are two agents (Point and Car) and three main tasks (Goal, Button, and Push), each with two levels of difficulty (1 and 2). Agents aim to reach the goal while avoiding any contact with obstacles. The environments are named using the following convention: {Agent}{Task}{Difficulty}. In the velocity-constrained environments, there are three agents: Ant, HalfCheetah, and Swimmer. The primary objective of these agents is to maximize

their rewards while adhering to the imposed velocity constraints. The environments are named in the convention of {Agent}Velocity.

**Bullet-Safety-Gym (Gronauer, 2022).** A suite of environments built upon the PyBullet physics simulator. These environments are similar to Safety-Gymnasium but feature a broader range of agents (including Ball, Car, Drone, and Ant). The tasks are relatively straightforward, with only two options available (Circle and Run). The environments are named through {Agent}{Task}.

### E.2. Experiment Settings

We train the baseline algorithms using the recommended hyperparameters specific to each task, for BCQ-Lag (Fujimoto et al., 2019; Stooke et al., 2020), CPQ (Xu et al., 2022a), COptiDICE (Lee et al., 2022), CDT (Liu et al., 2023b), TREBI (Lin et al., 2023), and FISOR (Zheng et al., 2024). To ensure a fair comparison, we train the baseline algorithms with three different random seeds and save the final output policy for safety evaluation. For each output policy, we conduct evaluations over 20 episodes to obtain reliable performance measures.

As for the training process of SDQC, we follow the neural network structure design and hyperparameter settings in Appendix C. Analogously, we select three different random seeds for training and perform evaluation over 20 episodes for each seed. To improve safety performance, we follow Zheng et al. (2024) to sample 16 candidate actions for each RL timestep, regardless of the safety assessment results and policy usage. The safest action is then selected based on the lowest $Q_h(z_{\theta_h}(s), a)$ value and executed as the final action.

### E.3. Computational Costs

We implement SDQC using PyTorch (Paszke et al., 2019) and conduct experiments on a single machine equipped with one GPU (NVIDIA RTX 4090, 24GB) and one CPU (AMD Ryzen 9 7950X). The training process comprises three phases. The first phase, known as the diffusion behavior cloner, demands approximately 1 hour for each task. For the second phase, which involves training representations and critics, the duration varies depending on the chosen network architecture. Attention-based architectures typically require over 4 hours, whereas MLP-based architectures typically demand around 1 hour. Finally, in the last training phase, the diffusion actor, convergence typically occurs in about 1 hour (without online testing).

Besides, we assess the inference time consumption of all baseline algorithms across 1000 RL timesteps on the CarPush2 task, averaging the results over 10 trials. Although SDQC is relatively slower compared to other non-autoregressive policies, it remains within an acceptable range.

*Table 8.* Inference time (seconds) comparison over 1000 RL timesteps.

| BCQ-Lag | CPQ | COptiDICE | CDT | TREBI | FISOR | SDQC |
|---------|-----|-----------|-----|-------|-------|------|
| 1.51 | 1.85 | 1.86 | 3.45 | 585.87 | 6.11 | 11.13 |

## F. Additional Experimental Results

### F.1. Addtional Generalization tests

In addition to the generalization tests (on the agent "Car") presented in Section 4.2, we perform generalization tests on the "PointGoal" and "PointPush" tasks (in Safety-Gymnasium), as illustrated in Figure 9. Similarly, the "Point" agent is challenged with tasks that involve reaching a goal point or pushing a box to a goal point in hazardous areas with obstacles, with the difficulty level varying between simple (PointGoal1, PointPush1) and challenging (PointGoal2, PointPush2).

Experimental observations reveal that, in comparison to the "Car" agent, the "Point" agent demonstrates a higher degree of inertia during its motion within the environment. Specifically, the "Point" agent lacks the ability to instantaneously halt or promptly alter its direction, thereby rendering the maintenance of safety more challenging in equivalent tasks when compared to the "Car" agent. This significantly undermines the generalization capability of most algorithms on the "Point" agent. For instance, the state-of-the-art (SOTA) safe offline RL algorithm FISOR performs poorly on the "Point" agent, exhibiting high costs across multiple environments. In contrast, our SDQC algorithm still achieves nearly zero violations in the majority of environments.

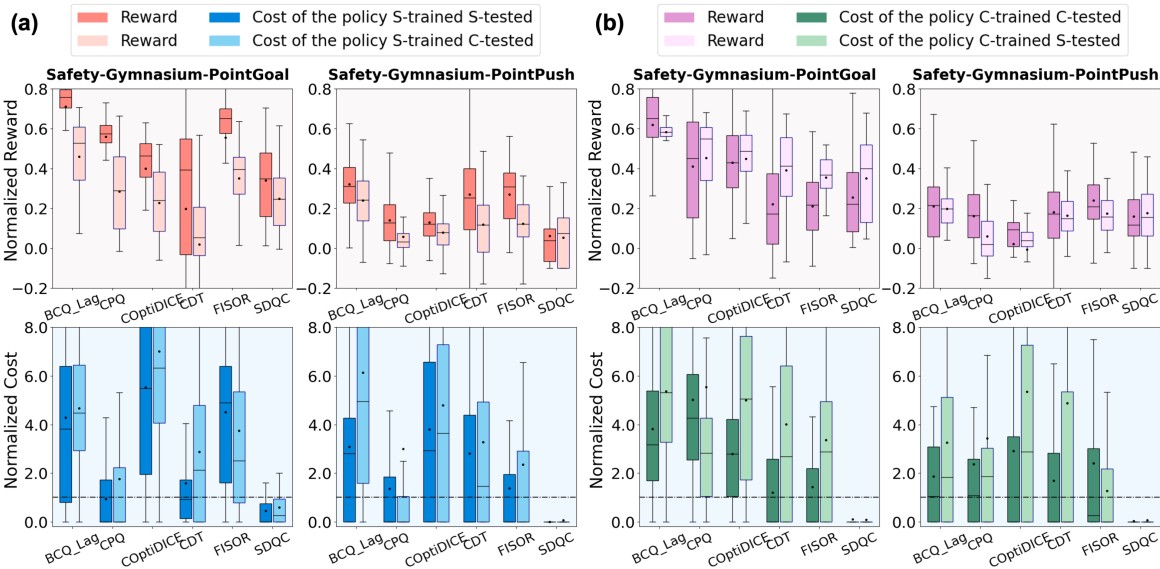

*Figure 9.* The generalization tests on the agent "Point" in Safety-Gymnasium.

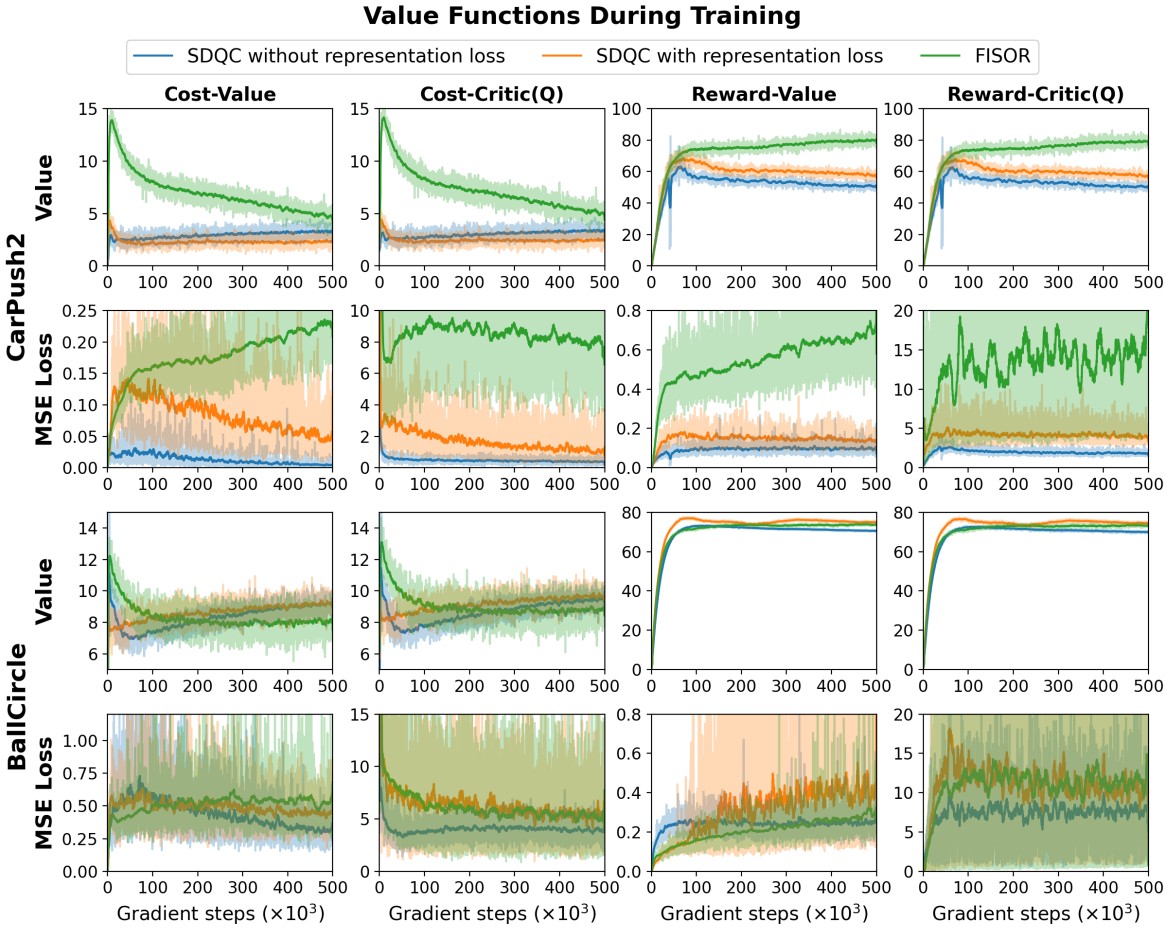

*Figure 10.* Comparison of value function loss during the training process among SDQC with and without representation loss, and FISOR, on the 'CarPush2' and 'BallCircle' tasks.

## F.2. Impact of Representation Loss on Value Estimations

A major concern for SDQC is that the joint optimization process of the value functions and representations (refer to Eqs. 8 and 12) may lead to instability in value estimation, subsequently affecting final performance. We analyze the performance of SDQC, both with and without representation loss, and compare it to FISOR by examining their respective Critic (Q) loss and Value loss patterns in relation to reward/cost metrics. This comparative analysis is conducted throughout the training process using two tasks: 'CarPush2' and 'BallCircle', with results illustrated in Figure 10. The experimental results indicate that while the inclusion of representation loss does lead to an increase in critic and value loss, it does not compromise the overall stability of the training. Furthermore, our proposed neural network architecture (used for "CarPush2"), with the incorporation of an attention-based state encoder, markedly improves the precision and stability of value function learning compared to the simple MLP utilized by FISOR.

## F.3. Learning Curves Comparison between SDQC and FISOR

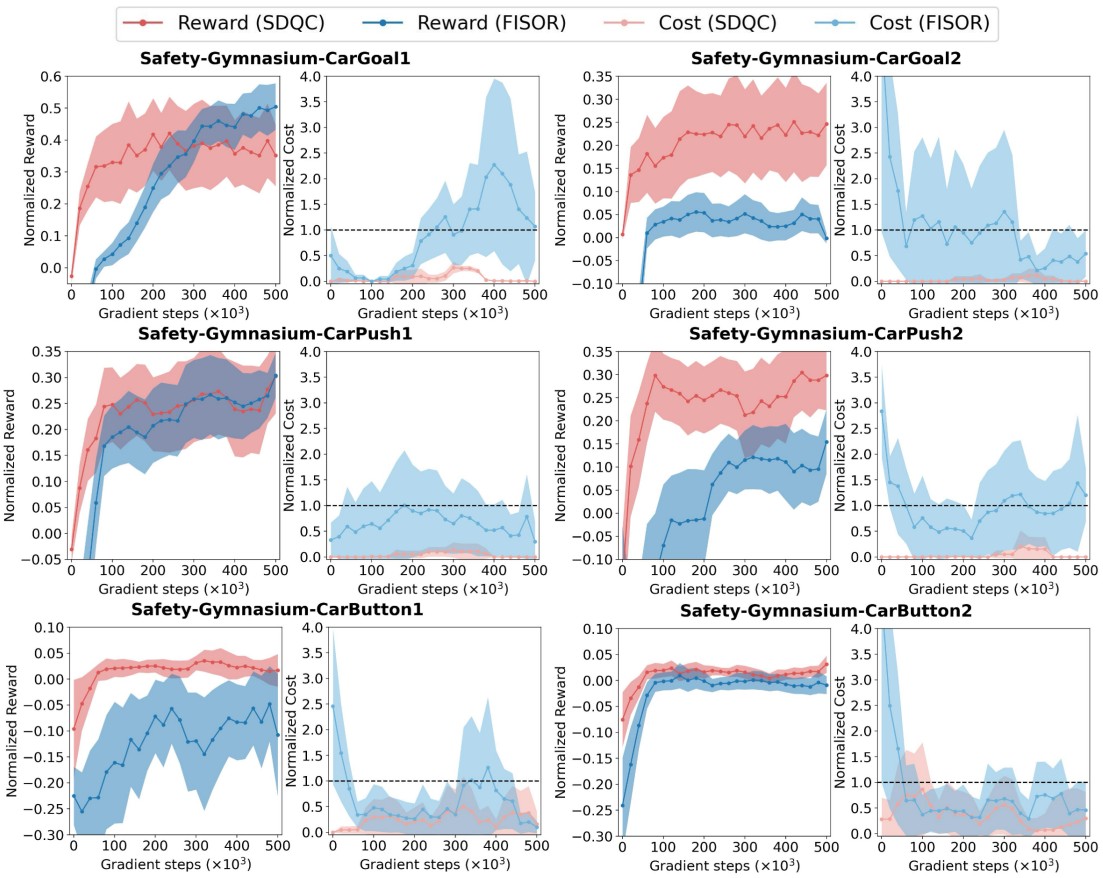

*Figure 11.* Training curves of SDQC and FISOR on the "Car" agent with tasks "Goal," "Push," and "Button" in the Safety-Gymnasium domain.

As complementary to the SOTA safe offline RL algorithm FISOR (Zheng et al., 2024), our SDQC employs the same implicit Q-learning method (Kostrikov et al., 2021) to learn optimal value functions and utilizes the safety Bellman operator in (Fisac et al., 2019) for safety assessment. Additionally, we adopt their approach for policy extraction through training a weighted regressed diffusion model. However, it should be noted that our decision-making process is based on decoupled representations rather than global observations. Furthermore, our policies are completely decoupled, and different policies are selected based on varying safety assessment results. To provide futher comparisons between SDQC and FISOR, we plot the training curves of both algorithms on the DSRL benchmark (Liu et al., 2023a) in Figure 11-14. The experimental results indicate that SDQC exhibits a higher level of safety assurance during training and achieves higher rewards in the majority of tasks.

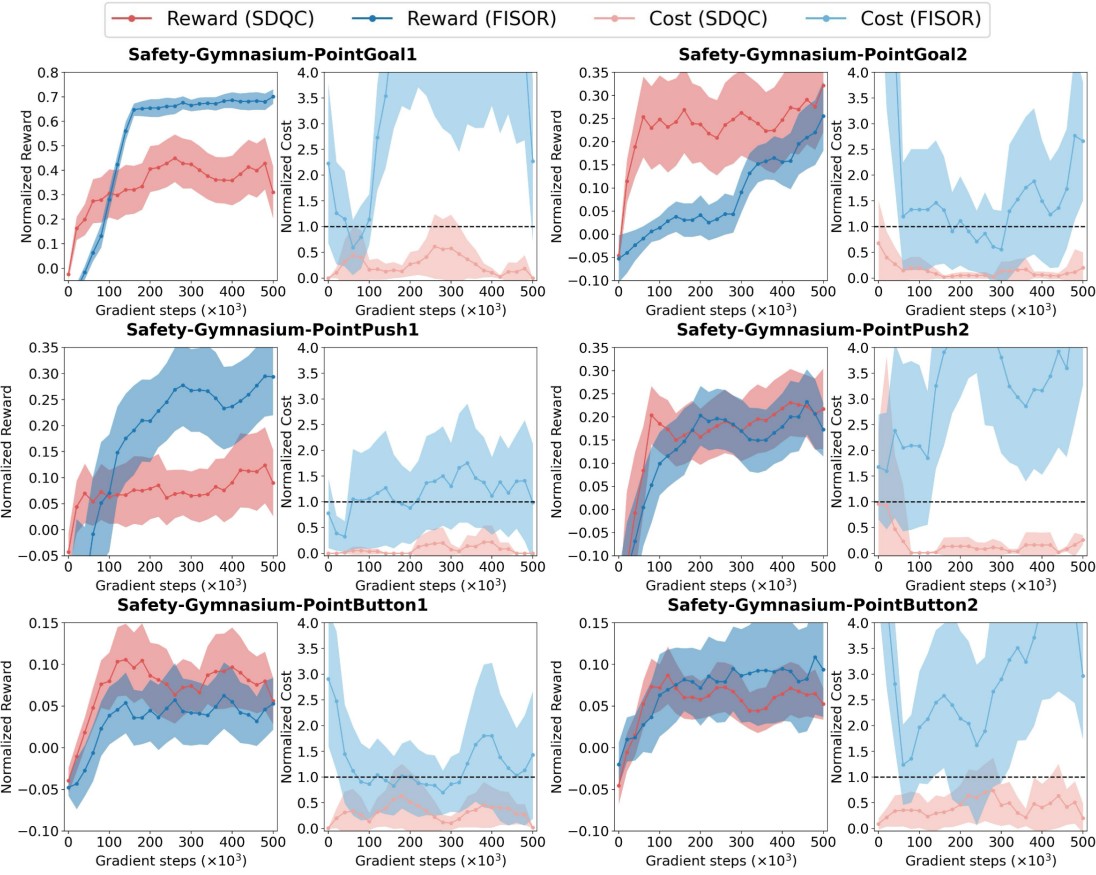

*Figure 12.* Training curves of SDQC and FISOR on the "Point" agent with tasks "Goal," "Push," and "Button" in the Safety-Gymnasium domain.

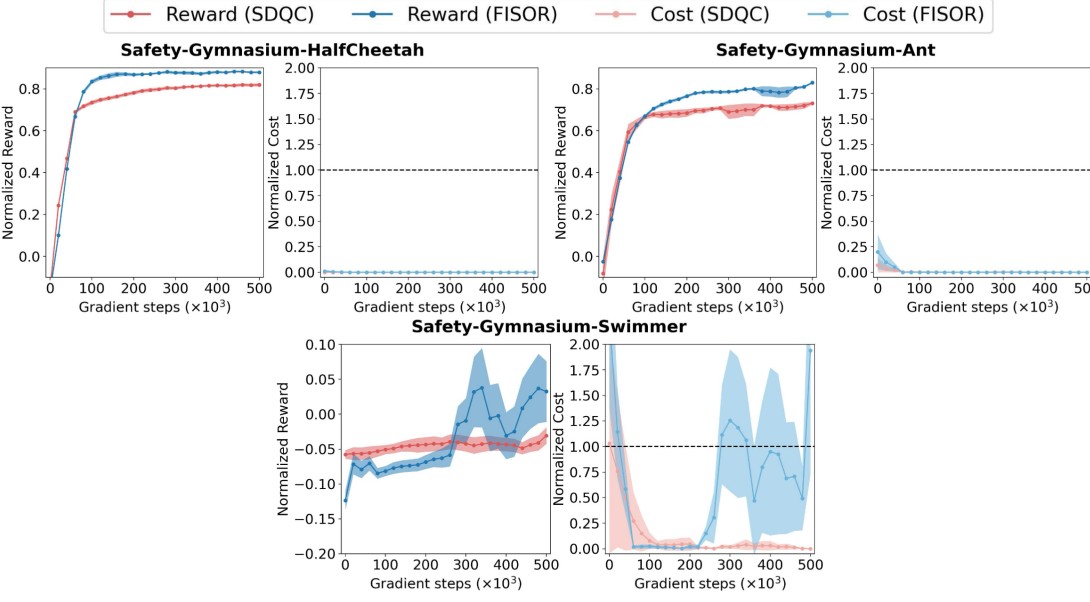

*Figure 13.* Training curves of SDQC and FISOR on the agents "HalfCheetah", "Ant", and "Swimmer" with velocity constraints task in the Safety-Gymnasium domain.

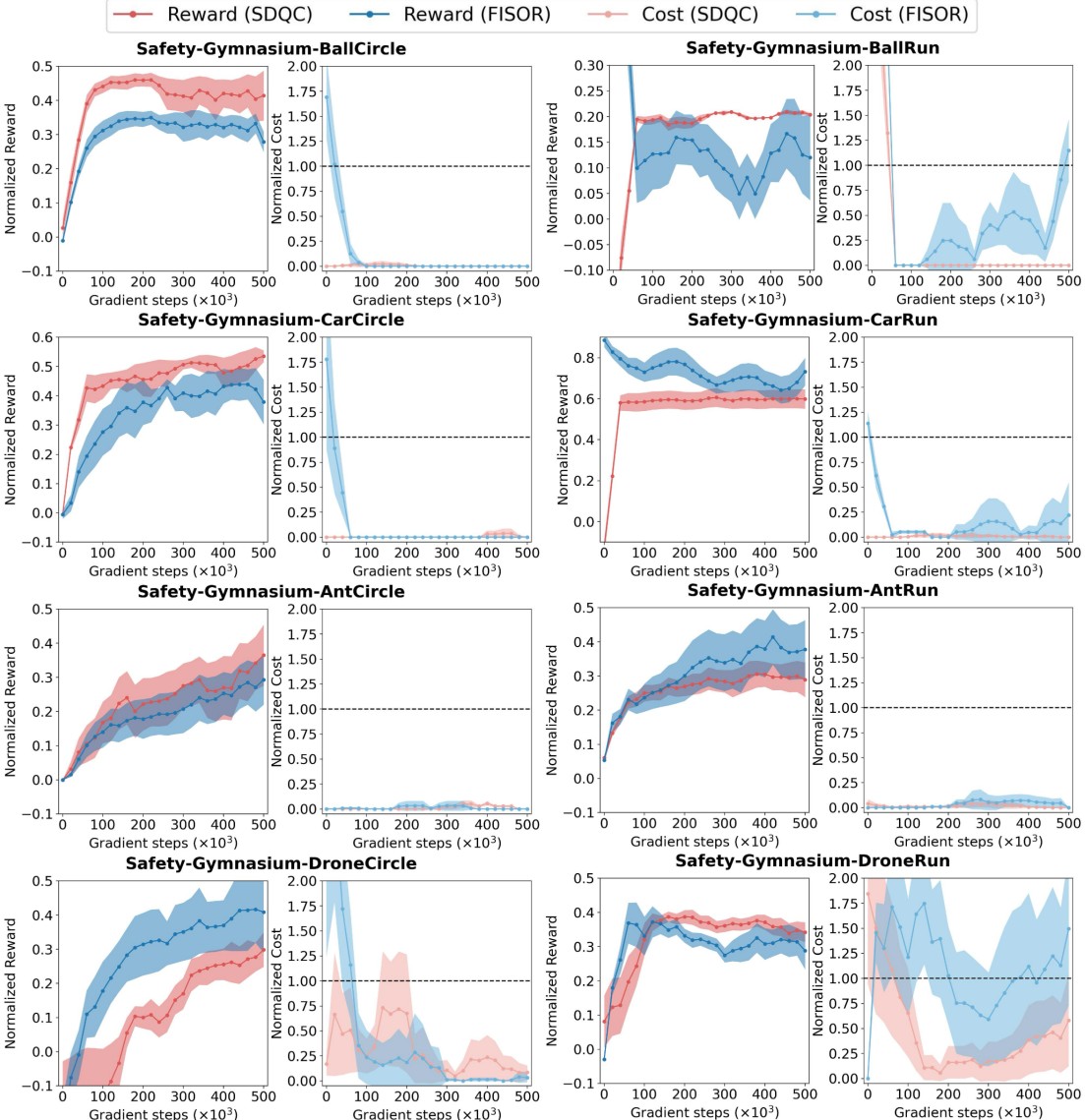

*Figure 14.* Training curves of SDQC and FISOR on the agents "Ball", "Car", "Ant", and "Drone" with tasks "Circle" and "Run" in the Bullet-Safety domain.

# G. Limitations and Future Works

One limitation of our current study arises from the substantial computational demands associated with training the SDQC model. This is particularly notable due to the necessity of executing three distinct training phases and the utilization of complex network architectures in certain scenarios. Despite this challenge, the remarkable cost-effectiveness and robustness to seed variance exhibited by our model mitigate these weaknesses. Looking ahead, our future research endeavors will prioritize the optimization of the training pipeline and the simplification of network structures to enhance training efficiency while maintaining performance standards.

