# OpenReview forum: "Q-Supervised Contrastive Representation: A State Decoupling Framework for Safe Offline Reinforcement Learning"
_ICML.cc/2025/Conference — ICML 2025 poster_

### Official Review · Reviewer_y1rM · 2025-02-27

**Overall Recommendation:** 4

**Summary:**

This paper introduces SDQC, a method designed to address the OOD problem in safe offline RL. By decoupling states into two independent representations, SDQC improves decision-making, safety, and generalization when facing unseen states. The method employs Q-supervised contrastive learning to guide the learning of state representations based on Q-values and selects policies based on safety evaluations. Experimental results show that SDQC outperforms baseline algorithms, achieving zero violations in most tasks. The paper also provides theoretical evidence that SDQC generates coarser representations while still preserving the optimal policy.

**Claims And Evidence:**

The claims in the paper are supported by clear and convincing evidence. This theoretical analysis supports SDQC’s ability to better handle OOD scenarios and perform well in unseen environments, which is validated through experimental results. Moreover, the theoretical analysis shows that SDQC can effectively separate these two aspects, allowing precise decisions under different safety conditions.

**Essential References Not Discussed:**

No.

**Experimental Designs Or Analyses:**

The comparative experiments with other baselines on the DSRL benchmark are sufficient, showing safer performance in all tasks.
The generalization experiments involved Goal and Push tasks of different difficulties, showing that SDQC achieves safer behavior and good rewards.
The ablation experiment is very thorough, involving not only the ablation of Q-supervised contrastive learning but also the ablation of contrastive learning parameters. The experiment fully demonstrates the impact of the contrastive learning module on the performance of the SDQC algorithm.

**Methods And Evaluation Criteria:**

The proposed method and evaluation standards are highly relevant to current problems and applications, particularly in the field of safe offline RL. The SDQC method addresses the challenge of OOD problems, which is crucial in safety-critical applications like autonomous driving. The use of the DSRL benchmark dataset allows comprehensive evaluation across various tasks, demonstrating SDQC's ability to achieve zero violations.

**Other Comments Or Suggestions:**

No.

**Other Strengths And Weaknesses:**

Strengths:
1. For the first time, Q-supervised contrastive learning is integrated into offline safe RL.
2. On the DSRL benchmark, comparative experiments on all tasks fully demonstrate the excellent security performance of the SDQC algorithm.
3. Ablation experiments on contrastive learning are sufficient and complete.

Weaknesses:
1. The generalization experiment is relatively simple, only dealing with Goal and Push tasks, without Button tasks, and does not vividly demonstrate the changes in task difficulty.

**Questions For Authors:**

1. Can provide other generalization experiments that are not Goal and Push tasks?
2. Can visualize scenarios of different difficulties of these tasks to illustrate the generalization performance?

**Relation To Broader Scientific Literature:**

This article innovatively combines contrastive learning with offline safe RL, which greatly improves the performance of existing algorithms and is of great significance to the development of offline safe RL algorithms.

**Theoretical Claims:**

The theoretical analysis is clear and the proof is rigorous. Theorem 3.1 highlights the relationship between bisimulation and Q*-irrelevance representations in MDPs, showing that the latter is a coarser but equally effective representation for deriving the optimal policy.

---

> ### Author Rebuttal · Authors · 2025-03-31
>
> We thank Reviewer y1rM for providing the positive feedback and recognizing the importance of our work. Please see our response to your questions below.
>
> ## (Weakness/Question1) More Generalization Experiments on "Button"
>
> We would like to clarify why we did not conduct generalization experiments on the Button task. As shown in Table 1 (page 6), all baseline algorithms exhibit extremely high costs on Button-related tasks. While our SDQC framework achieves safe results, the corresponding reward remains exceptionally low.
>
> In the Button task, the agent is required to reach the orange button (marked with a green cylinder) without colliding with obstacles, stepping into traps, or activating incorrect buttons. At difficulty level 2, however, the environment includes 17 distractors, six of which are dynamic and move around the agent. (For a visualization, please refer to the results [**[here]**](https://github.com/wrk8/SDQC-Rebuttal/blob/main/to_reviewer_y1rm/CarButton.gif).)
>
> The optimal policy learned by the agent trained with SDQC involves moving alongside the dynamic obstacles to avoid collisions, but this behavior largely ignores the target button. In contrast, agents trained with other baseline algorithms tend to disregard the moving obstacles entirely and recklessly navigate toward the target button. These completely contradictory behaviors make comparisons between algorithms meaningless.
> Conversely, in other tasks, such as Goal and Push, agents trained with different algorithms exhibit similar behaviors, enabling fair generalization tests in these environments. (For reference, additional generalization experiments on the "PointGoal" and "PointPush" tasks are provided in Figure 9 (page 25) of our main paper.)
>
> We will include a more detailed discussion of this phenomenon in the final version of the paper.
>
> ## (Question2) Visualization results of difficulty-level.
>
> Thank you for your insightful comments. We agree that clear visualizations can better illustrate the generalization performance of our SDQC.
>
> In the Goal-related tasks, the agent is required to reach the goal (marked with a green cylinder) while avoiding traps and collisions with obstacles. A visualization of this task is provided [**[here]**](https://github.com/wrk8/SDQC-Rebuttal/blob/main/to_reviewer_y1rm/PointGoal.gif). At difficulty level 1 (row 1 in the GIF), the environment contains 8 traps and 1 obstacle. At difficulty level 2 (row 2), the complexity increases, with 10 traps and 10 obstacles.
>
> In the Push-related tasks, the agent is required to push a yellow box to the goal (marked with a green cylinder) while avoiding traps and collisions with solid pillars. A visualization of this task is available [**[here]**](https://github.com/wrk8/SDQC-Rebuttal/blob/main/to_reviewer_y1rm/CarPush.gif). At difficulty level 1 (row 1 in the GIF), the environment consists of 2 traps and 1 pillar. At difficulty level 2 (row 2), the environment becomes more challenging, with 4 traps and 4 pillars.

---

### Official Review · Reviewer_JBF8 · 2025-03-13

**Overall Recommendation:** 2

**Summary:**

This paper proposes a new safe offline RL algorithm called SDQC, the key idea of the algorithm is to decouple observations into reward and cost-related representations, and use them in a value & policy learning framework similar to the one proposed in FISOR. The key intuition is that the values related to reward and cost may depend on different state representations, and decoupling them can help enhance generalization and safety. The proposed method achieves impressive performance in the DSRL benchmark. However, the algorithm is also quite heavy, and needs to learn two distinct state representations plus three different diffusion policy models.

**Claims And Evidence:**

The paper claims that it can achieve strong safety and generalization performances, which are well-supported by the DSRL benchmark results and additional generalization tests.

**Essential References Not Discussed:**

N/A

**Experimental Designs Or Analyses:**

The evaluation is generally reasonable, but there are also some aspects that can be further improved:
- SDQC utilizes **two sets of representations, three diffusion policies, and transformer-based (ATN) encoder** during training, making it very complex and computationally expensive. The authors should compare SDQC with baselines with similar scales of parameters, training time, and inference time. Additionally, given the similarity between FISOR and SDQC, it would be valuable to evaluate FISOR's performance using the same transformer encoder and separated diffusion policy model. Furthermore, testing FISOR with representation learning could help isolate and highlight the contribution of state-decoupled representation learning.
- Adding more ablation and discussion on temperature hyperparameters $\iota_r, \iota_h, \iota_{to}$ in Eq. (14).

**Methods And Evaluation Criteria:**

**Regarding Method:**
- In Equation 5, the contrastive loss includes two exponential terms, $exp(s, \tilde{s})$ and $exp(z, \tilde{z})$, which might lead to instability during training. Since training is coupled with the value function update, this instability could be further amplified, especially in applications using image inputs.
- The authors use only a subset of the dataset for training, whereas contrastive learning typically requires a large amount of training data. It is unclear why using a limited number of anchor states would lead to better performance. Clarification on this point would be helpful.
- The authors introduce an upper-bound cost value function but lack sufficient theoretical analysis to explain its significance. Since this function is used to identify safe conditions, further clarification is needed.

**Regarding Evaluation:**
- Using the DSRL benchmark and additional generalization tests is reasonable.

**Other Comments Or Suggestions:**

N/A

**Other Strengths And Weaknesses:**

**Strengths:**
- SDQC focuses on the hard constraint setting and achieves state-of-the-art performance.

**Weaknesses:**
- See my above comments in the Methods, Theoretical Claims, and Experimental Designs sections.

**Questions For Authors:**

In Table 4, different parameter settings are used for different tasks. How does the performance compare when using the same parameters across all tasks?

**Relation To Broader Scientific Literature:**

Safe offline RL methods are relevant to industrial control, robotics, and autonomous driving.

**Theoretical Claims:**

I do not think the theoretical analysis in Section 3.4 has much relevance to the proposed algorithm. Theorem 3.3 only discusses the properties of bisimulation representations and $Q^*$-irrelevance representation. The proposed contrastive representation has nothing to do with bisimulation representation, and there is also no formal proof or discussion given to show the proposed representation is a $Q^*$-irrelevance representation. Hence Theorem 3.3 is completely irrelevant. The authors then use Theorem 3.3 and the entropy maximization design (Eq.(4)) to claim the proposed method has better generalization (has larger conditioned entropy), which in my opinion, is seriously flawed. First, whether Theorem 3.3 holds for your representation is questionable; second, why does larger conditioned entropy justify better generalization?

I suggest removing Section 3.4 and the "theoretical analysis" entirely from the paper, as it is irrelevant and misleading, and cannot provide convincing arguments to demonstrate the superiority of the proposed method. If the authors really want to keep the theoretical analysis in the paper, at least the authors should:
- Rigorously prove the proposed contrastive representation is a $Q^*$-irrelevance representation, which I personally believe is impossible, as contrastive representations typically don't have many good properties for decision-making problems.
- Use a more rigorous generalization metric for discussion, instead of using the magnitude of conditioned entropy.
- Lastly, even if the current logic of the theoretical analysis holds (if both the above issues can be proved), the authors can only show their method is better than bisimulation representations, which is a rather weak conclusion and does not offer much real-world relevance.

---

> ### Author Rebuttal · Authors · 2025-03-31
>
> We thank Reviewer JBF8 for the valuable suggestions. We would like to address your concerns point by point below.
> ## (Method1) Potential instability for Eq. 5
> In Eq. 5, the exponential coefficient $\Gamma(s,\tilde{s})$ is a detached soft measure introduced in [1] that helps stabilize training.
> Regarding the potential instability caused by coupling value function updates with representation learning, a detailed analysis is provided in Appendix F.2 (pages 25–26).
> Experimental results demonstrate that the contrastive term has no adverse effect on value estimation, while our proposed ATN-based representation network significantly reduces estimation error.
> ## (Method2) Why limited anchors lead to better results
> We apologize for the unclear description. We use the full dataset for training, not just a subset. For each gradient step, we sample a batch of data and perform contrastive learning within the batch.
> Since our total loss includes multiple components, using more anchors could cause the neural network to overly focus on learning the representation, which might result in suboptimal performance of the value function.
> ## (Method3) Upper-bound of cost value function
> Please note that $Q_{h}$ is updated based on the TD error computed relative to $V_{low}$ rather than $V_{up}$. In contrast, $V_{up}$ is updated toward $Q_h$ via expectile regression parameterized by $\tau$. According to Lemma. 1 in [2], it is straightforward that $\lim_{\tau\rightarrow 1}V_{up}(z(s))=\max_{a\in\mathcal{A}_\beta^s}Q_h(z(s),a)$.
> ## (Theory1) Why introduce a comparison with "bisimulation"
> We conduct a theoretical comparison with bisimulation, which is one of the most widely used representation learning methods in RL [1, 3]. While implementing our idea of decoupling states for decision-making, bisimulation initially seemed like a natural choice.
> However, this approach proved unsuccessful. It fails during the initial model-estimation stage due to the non-smooth nature of the cost function and its sparse value distribution. Similar challenges have been observed in sparse reward RL tasks [4].
> In contrast, our SDQC method avoids these issues by using smooth Q-values and provides stronger theoretical advantages. We will include this failed attempt in the final version of our paper.
> ## (Theory2) Contrastive representations do not have good properties for decision-making.
> In fact, contrastive-based representations are widely used in RL [1]. Similar to previous studies, our analysis is built upon the MDP framework, while our simulations address high-dimensional, infinite state and action spaces.
> We do not claim that contrastive learning achieves a rigorous Q-irrelevance representation; indeed, such a result is unattainable for infinite problems approximated by neural networks. Rather, we cluster the Q-indicated representations in high-dimensional space, analogous to previous works that cluster representations with their associated indicators.
> ## (Theory3) Why conditional entropy can indicate "generalization ability"
> Due to space limitations, please refer to our response to Reviewer zA7P on **Explanations of "conditional entropy"**.
> ## (Experiment1) "Fair comparison" with baselines.
> The required "fair comparison based on same computational cost" is hard to execute.
> First, each baseline algorithm has unique optimal parameter settings suggested by their authors.
> Moreover, some baselines, such as CDT with GPT-2 backbone and TREBI with U-Net structure, are even more computational expensive than our SDQC.
>
> According to reviewer's requirement, we implement FISOR with ATN-based structure and separate diffusion policies [**[here]**](https://github.com/wrk8/SDQC-Rebuttal/blob/main/to_reviewer_JBF8/compare_FISOR_SDQC.png).
> ## (Experiment2) Temperature ablations for diffusion policies
> Please find ablations [**[here]**](https://github.com/wrk8/SDQC-Rebuttal/blob/main/to_reviewer_JBF8/Ablation_diffusion_temp.png). The default parameters follow FISOR and are kept consistent across all environments.
> ## (Question) Consistent hyperparameters for all environments
> We would like to emphasize that our choice of hyperparameters is domain-specific.
> (We use the same hyperparameters across 12 tasks in safety-gym domain.)
> Please refer to our response to Reviewer zA7P on "Complex hyperparameter settings".
> We argue that **no RL algorithm can employ a unified set of hyperparameters across all domains and environments**, including most commonly used SAC and PPO.
>
> [1] Agarwal, Rishabh, et al. Contrastive behavioral similarity embeddings for generalization in reinforcement learning. ICLR (2021).
>
> [2] Kostrikov, Ilya, et al. Offline Reinforcement Learning with Implicit Q-Learning. ICLR (2021).
>
> [3] Castro, Pablo. Scalable methods for computing state similarity in deterministic markov decision processes. AAAI (2020).
>
> [4] Lee, Vint, et al. DreamSmooth: Improving Model-based Reinforcement Learning via Reward Smoothing. ICLR (2024).

---

> > ### Comment · Reviewer_JBF8 · 2025-04-08
> >
> > I'd like to thank the authors for the response. However, many of my concerns remain. Specifically,
> > - **Regarding theoretical analysis**: I'm not asking why introduce a comparison with bisimulation, but saying the theoretical analysis in Section 4.3 is not relevant to your proposed method. As I have mentioned in my comment, Theorem 3.3 only discusses the properties of bisimulation representations and $Q^*$-irrelevance representation. As your algorithm does not learn bisimulation representations, and the authors cannot prove their representations are $Q^*$-irrelevance representations, thus the entire theoretical analysis has nothing to do with the proposed algorithm. The authors are dodging my comment.
> > - **Regarding why conditional entropy can indicate "generalization ability"**: The authors have added some explanation on conditional entropy, but this still doesn't provide any concrete evidence/theoretical support on "why this can justify better generalization".
> > - **Heavy and complex algorithm**: As I have stated, "SDQC utilizes two sets of representations, three diffusion policies, and transformer-based (ATN) encoder during training", given such complexity and extra hyperparameters, few people would want to use it to solve real-world tasks. There are simply too many components to be tuned to make it work.
> > - **Hyperparameters**: In fact, it is recommended in offline RL to use consistent hyperparameters for different tasks, since in real-world deployment, it would be infeasible or even dangerous for model/hyperparameter selection on real-world systems. Keeping a minimal set of hyperparameters or having hyperparameter robustness is extremely important for an offline RL algorithm to be useful. The authors use SAC and PPO as an example, which is not a meaningful argument, since in online RL learning in the simulation environment, for sure, you can perform unrestricted hyperparameter tuning, but it is simply not the case for offline RL scenarios. Also, it should be noted that the baseline FISOR uses the same set of hyperparameters for all of its experiments.

---

> > > ### Author Response · Authors · 2025-04-08
> > >
> > > We sincerely acknowledge reviewer JBF8 to provide further feedback. To address your remaining concerns, please refer to our responses below.
> > >
> > > ## Regarding theoretical analysis
> > >
> > > We would like to emphasize that we are NOT dodging your comments.
> > >
> > > Firstly, you requested a rigorous proof that our Q-supervised contrastive representation is a $Q^{\*}$-irrelevant representation. As we have clarified, such a result is unattainable for infinite problems approximated by neural networks. However, similar to prior work [1], we assume that contrastive learning, when guided by appropriate indicators, can perfectly cluster the representations and achieve the ideal effect. The theoretical analysis is then conducted for this idealized setting.
> > >
> > > Secondly, you are concerned that the comparison between bisimulation and $Q^{\*}$-irrelevance representation is not relevant to our methods.
> > > In response, we claim that we tried both methods (through contrastive learning with assumption that the representations are ideally clustered). While bisimulation initially appeared to be a mature and natural approach, it failed due to the non-smooth nature of the cost function. In contrast, our SDQC method avoids these issues by using smooth Q-values and provides stronger theoretical advantages.
> > >
> > > ## Regarding why conditional entropy can indicate "generalization ability"
> > >
> > > Here, we would like to reiterate the step-by-step reasoning that connects "generalization ability" to "conditional entropy":
> > > 1. According to [2], coarser representation in RL generally leads to improved generalization performance due to reduced state space.
> > > 2. To obtain coarser representations, we treat the representation as a random variable and aim to minimize its information content, i.e., the entropy $H(z)$.
> > > 3. Minimizing $H(z)$ is equivalent to maximizing $H(s|z)$, as the relationship $H(s|z) = H(s) - H(z)$ holds, and $H(s)$ is a constant for a fixed offline dataset.
> > >
> > > ## Heavy and complex algorithm
> > >
> > > We acknowledge that our algorithm has a more complex structure compared to existing ones, and we have explicitly discussed this as a limitation in Appendix G.
> > > However, we believe that our primary contribution lies in the introduction of the concept of "decoupling states for decision-making", which has been shown to be highly effective in our experiments. While there may still be potential room for simplifying the current framework, we believe it provides valuable insights for future research.
> > >
> > > ## Unified Hyperparameters
> > >
> > > We agree with your perspective that offline RL algorithms, in theory, should impose stricter requirements on hyperparameter consistency compared to online RL.
> > > However, this is often impractical in most cases. For instance, two well-known offline RL algorithms, MOPO [3] and Diffusion-QL [4], require different hyperparameter configurations across various environments to achieve satisfactory performance.
> > > As you pointed out, FISOR [5] performs well in this regard by employing a unified hyperparameter setting across all environments. Nonetheless, it fails to achieve strictly zero-cost (its original target) in over 70% of environments.
> > >
> > > In our work, we emphasize that SDQC requires different hyperparameter configurations across environments primarily due to its reliance on representation learning. Unlike image-based RL tasks, state-based RL tasks typically involve observations with high information density. Over-compression or over-expansion of these observations in the representation space can result in the collapse of the final outcomes. Thus, **it is necessary to adjust the dimension of the representation space according to the original dimensionality of the observations**.
> > >
> > > As shown in Table 3 of our main paper, we have ensured that **most hyperparameters remain consistent across all environments and domains**. Users only need to select the network architecture and adjust the encoded state dimension based on the global observation space, which is largely determined by the physical characteristics of each domain. Additionally, we provide detailed guidelines for these selections (see Appendix C.3, lines 1097–1133). We do NOT consider such hyperparameter configuration approach to be unacceptable.
> > >
> > > [1] Agarwal, Rishabh, et al. Contrastive behavioral similarity embeddings for generalization in reinforcement learning. ICLR (2021).
> > >
> > > [2] Liu, Guoqing, et al. Return-based contrastive representation learning for reinforcement learning. ICLR (2021).
> > >
> > > [3] Yu, Tianhe, et al. MOPO: Model-based offline policy optimization. NeurIPS (2020).
> > >
> > > [4] Wang, Zhendong, et al. Diffusion policies as an expressive policy class for offline reinforcement learning. ICLR (2023).
> > >
> > > [5] Zheng, Yinan, et al. Safe Offline Reinforcement Learning with Feasibility-Guided Diffusion Model. ICLR (2024).

---

### Official Review · Reviewer_zA7P · 2025-03-13

**Overall Recommendation:** 4

**Summary:**

This paper introduces a framework "State Decoupling with Q-Supervised Contrastive Representation" (SDQC) to improve safe offline reinforcement learning. They do this by using two distinct state representations (the "decoupling"), one being for rewards and one for costs, and learn these using contrastive objectives. They then use these representations to extract three policies: a "reward policy", "cost policy", and "tradeoff policy", which they train using diffusion models. In practice, they use a safety assessment function on the two representations of a given observation, which dictates which policy is used at that timestep. They evaluate their method against a number of baselines across the DSRL benchmark, as well as further experiments to compare generalization capabilities of different algorithms.


---

Not sure where I'm supposed to acknowledge the rebuttal so I guess I'll do it here.

I thank the authors for a strong rebuttal, it cleared up the uncertainties I had and I will adjust my score to an accept. One thing I will note is that I'm not a fan of the term ``reverse expectile regression'' in its use as the authors described, but if it's used in prior work I can't fault them for it.

**Claims And Evidence:**

I believe that the claims made are mostly supported by clear and convincing evidence (I have a couple questions on some theoretical aspects below, but I believe these should be squashable in the rebuttal).

**Essential References Not Discussed:**

The authors state that their method is a useful alternative to bisimulation as it doesn't require model-based methods in order to estimate. However there is a line of work on estimating bisimulation-type distances from samples in a model-free setting, for example beginning with [1] from NeurIPS 2021 and a number of works which followed.

[1] Pablo Samuel Castro, Tyler Kastner, Prakash Panangaden, and Mark Rowland. MICo: Learning improved representations via sampling-based state similarity for Markov decision processes. In Advances in Neural Information Processing Systems, 2021.

**Experimental Designs Or Analyses:**

Their experiments on the DSRL benchmark appears sound to me (although no code was provided).

**Methods And Evaluation Criteria:**

I believe the proposed methods and evaluation criteria make sense.

**Other Comments Or Suggestions:**

- I think the cross-entropy $H(s| z_\theta(s))$ should be defined prior to its use in Equation (4). This definition confuses me, as cross entropy is normally defined with respect to two probability distributions, but here your arguments are a state and the state's representation.
- Following the above: the way $H(s| z_1)$ is used in the proof of Proposition A.4. seems to be a variation of the KL divergence rather than cross entropy.
- It is stated a number of times that a coarser representation has better generalization properties. Perhaps briefly arguing why this might be the case, or citing references that go into detail on this fact, can be helpful.

Minor typos/nits:
- Lemma A.3: "finner" -> "finer".
- For quotation marks, use ``<>'' instead of ''<>'' (otherwise the opening one is reversed).
- A representation being finer than another is defined but a representation being coarser than another is used without definition.

**Other Strengths And Weaknesses:**

**Strengths**
- The method introduced appears novel and non-incremental (at least to my knowledge), and obtains strong results against baselines on the environments considered.
- I find Section C of the appendix to be really great to help the reader understand and build intuition for the algorithm (pseudocode + architecture diagrams).
- The GIF in the supplementary material is super helpful to build intuition and understand the 3 policies working together (this could've been advertised to the reviewers better! I came across this accidentally).


**Weaknesses**
- The proposed algorithm seems quite complex with a number of moving parts, which may make it brittle with respect to hyperparameter choices. (I appreciate that the authors demonstrate that this isn't the case with respect to the diffusion hyperparameters at least).
- Similar to the above, I can imagine the number of moving parts may make it run potentially slow (multiple diffusion models, nearest neighbour search, contrastive objective, etc).
- Some of the theory presented in Section 3.4. feels a bit hard to follow for me and I do not entirely see the motivation (I don't "type-check" the variant of cross-entropy they use, and I also don't see how this relates to generalization). I expand on this in my comments and questions in the sections below.

**Questions For Authors:**

- What is "reverse expectile regression" in **Cost-related Representation** of Section 3? It seems like the standard expectile loss?
- Following my second weakness: how does the wall-clock time of your method compare to alternative methods?
- In Table 1, what is a "Safe agent" vs a "Unsafe agent"?
- Following my comments in the previous section, what definition of cross-entropy are you using in the terms $H(s|z)$?

**Relation To Broader Scientific Literature:**

This paper is at the intersection of safe RL, offline RL, and representation learning in RL. It borrows some techniques from previous work in safe offline RL (such as the use of expectile losses, using regressed diffusion models to train the policies, and using a diffusion behaviour cloner. They primarily build on this work by improving representation learning in a novel way.

**Theoretical Claims:**

I checked all proofs and theoretical claims made. I had some questions which I highlighted in the sections below.

---

> ### Author Rebuttal · Authors · 2025-03-31
>
> We acknowledge Reviewer zA7P for appreciating the significance and novelty of our work. Below, we respond to the your questions point by point.
>
> ## (References) Missed discussing on relevant references
>
> The suggested papers are indeed highly relevant to our work. They represent extensions of bisimulation-series representation methods. We will incorporate a discussion of these papers in Section 3.4 and the Related Works section.
>
> ## (Weakness1) Complex hyperparameter settings
>
> We aknowledge SDQC involves multi-step training and many hyperparameters. However, as shown in Table 3, we have set most of the hyperparameters to be consistent across all environments and domains. Users only need to select the network structure and adjust the encoded state dimension according to the global observation space, which is largely determined by the physical characteristics of each domain. In fact, we provide detailed guidelines for selecting them (in Appendix C.3, lines 1097-1133).
>
> ## (Weakness2/Question2) Computational costs
>
> The computational costs for SDQC are indeed slight higher than other baseline algorithms. We discussed this point in Appendix E.3, lines 1288-1302.
> To provide a clear comparison, we implement all algorithms with pytorch on a single machine equipped with one GPU (NVIDIA RTX 4090) and one CPU (AMD Ryzen 7950X), and report the results on *CarPush2* task.
> During the training phase, the total training time (hours) are presented as follows.
>
> |BCQ-Lag|CPQ|COptiDICE|CDT|TREBI|FISOR|SDQC(ours)|
> |-|-|-|-|-|-|-|
> |0.46|0.45|0.41|3.12|16.62|1.77|5.43|
>
> During the testing (inference) phase, we measure the time consuption (seconds) over 1000 RL timesteps as follows.
>
> |BCQ-Lag|CPQ|COptiDICE|CDT|TREBI|FISOR|SDQC(ours)|
> |-|-|-|-|-|-|-|
> |1.51|1.85|1.86|3.45|585.87|6.11|11.13|
>
> ## (Weakness3/Question4) Explanations on "conditional entropy"
>
> We apologize for any conceptual confusion caused by the missing definition. Here, we clarify that we denote $H(s|z)$ as **conditional entropy**, which is used measure information content of random variables, rather than cross-entropy or KL divergence, which are used to measure the distance between prob-distributions.
> Conditional entropy is a concept in information theory, and its formal definition was introduced by Shannon[1]:
>
> $H(s|z)=-\sum_{s,z} p(s,z) \log p(s|z)$
>
> where $p$ represents the probability function.
> According to [2], coarser representation in RL generally introduce better generalization performance due to reduced state space. Thus, when treating the representation as a random variable $z$, **we initially aim to minimize the information content of $z$, i.e., the entropy $H(z)$**. Note that the following relationship always holds:
>
> $H(s|z) = H(s,z) - H(z) = H(s) - H(z)$
>
> where the second equality is valid since $z$ is a function of $s$. For any given offline dataset, $H(s)$ is fixed. Therefore, **minimizing $H(z)$ is equivalent to maximizing $H(s|z)$**.
>
> We will add above detailed explanation in the final version of our paper.
>
> ## (Typos)
>
> Thanks for pointing out these! We have carefully checked entire paper to correct all typos.
>
> ## (Question1) What is "reverse expectile regression"
>
> The optimal value function under the general Bellman operator is given by $V_r(s)=\max_a Q_r^*(s,a)$, and is $V_h(s)=\min_a Q_h^*(s,a)$ under the cost-related safe Bellman operator. The expectile regression $L^{\tau}(u)=\vert \tau - \mathbb{I}(u<0)\vert u^2$, with $\tau \in (0.5, 1)$ can be used to approximate the maximum value. Conversely, the reverse version $L^{rev}_{\tau}(u)=\vert \tau - \mathbb{I}(u>0)\vert u^2$ with $\tau \in (0.5, 1)$ can be utilized to approximate the minimum value. The difference lies in the direction of the indicator function $\mathbb{I}(u)$. Following the setting in FISOR [3], we refer to this as "reverse expectile regression".
>
> ## (Question3) What is Safe Agents and Unsafe Agents in Table 1
>
> As introduced in Section 4.1 (lines 320–325, column 1, and lines 299–303, column 2), our ultimate objective is to achieve zero cost during testing, in alignment with the framework established by FISOR [3]. However, most baseline algorithms struggle to perform effectively under a zero-cost threshold. Therefore, following FISOR [3], we impose a strict cost limit of 10 for the Safety-Gymnasium environment and 5 for the Bullet-Safety-Gym environment. Agents with an **average cost below this threshold** during testing are classified as **safe agents**, while those **exceeding the threshold** are classified as **unsafe agents**.
>
> **Lastly, and importantly, we are committed to open-sourcing all code upon acceptance.**
>
> [1] Shannon, C. E. A mathematical theory of communication. The Bell system technical journal (1948).
>
> [2] Liu, Guoqing, et al. Return-based contrastive representation learning for reinforcement learning. ICLR (2021).
>
> [3] Zheng, Yinan, et al. Safe Offline Reinforcement Learning with Feasibility-Guided Diffusion Model. ICLR (2024).

---

> > ### Comment · Reviewer_zA7P · 2025-04-03
> >
> > I initially appended this to my original review as I didn't see the rebuttal comment option, but I'll add this here again for posterity.
> >
> > I thank the authors for their rebuttal, it cleared up the uncertainties I had and I will adjust my score to an accept. One thing I will note is that I'm not a fan of the term ``reverse expectile regression'' in its use as the authors described, but if it's used in prior work I can't fault them for it.

---

> > > ### Author Response · Authors · 2025-04-04
> > >
> > > We sincerely thank reviewer zA7P for acknowledging our rebuttal and for improving the score. Regarding your final concern about the naming convention of ``reverse expectile regression", we agree that the terminology is not entirely rigorous.
> > >
> > > After carefully reviewing the pioneering work IQL [1], we note that they refer to expectile regression used to approximate the maximum value as **upper expectile regression**. Following this convention, we will rename the version used to approximate the minimum value as **lower expectile regression**. We will rewrite the corresponding parts in the final version of our paper. Thanks again for your feedback!
> > >
> > > [1] Kostrikov, Ilya, et al. Offline Reinforcement Learning with Implicit Q-Learning. ICLR (2022).

---

### Official Review · Reviewer_cp6m · 2025-03-14

**Overall Recommendation:** 4

**Summary:**

This paper introduces State Decoupling with Q-supervised Contrastive representation (SDQC), a framework that decouples the global observations of an RL agent into reward- and cost-related representations to deal with OOD data and improve generalisation.

**Claims And Evidence:**

1) The authors claim that "Safe offline RL [...] provides a promising solution that learns the safety-guaranteed policy in a fully offline manner. Its training requires no risky interaction with the environment and relies only on the precollected offline dataset."

I am not familiar with offline safe RL, but it looks like the problem has only been moved, since how is the precollected offline dataset of execution going to be collected?

2) "The representations solely capture either reward or cost information, independent of another factor, as determined by the training of Q*."

How is it possible to decouple this information determined by the training of Q*?

3) The authors claim superior experimental performance, which looks backed by empirical evidence.

4) On p. 5 the authors say that "our Q-supervised contrastive learning method theoretically surpasses bisimulation in terms of generalization."
But what it is meant by generalisation at this point? So far generalization has only been discussed informally in the introduction.

**Essential References Not Discussed:**

It doesn't look like.

**Experimental Designs Or Analyses:**

The experiments look fine in terms of baselines and benchmarks for comparison.

**Methods And Evaluation Criteria:**

The empirical evaluation is rather thorough. It considers 5 other relevant methods in the benchmark DSRL.

As states above, I am not sure that offline learning is the best solution for safety in RL, as the agent has to interact with an unsafe environment to collect experiences.

**Other Comments Or Suggestions:**

1) The example in the introduction does not really add much to the discussion, as I'd assume that most people are familiar with issues with OOD data and generalisation.

**Other Strengths And Weaknesses:**

1) The authors stress the relevance of OOD generalization for their contribution, but then generalization is only discussed in Sec. 4.2, rather marginally.
I wonder whether a stronger case for the paper can be made by simply considering the improvement in performance.

2) The authors manage to fit safe (offline) learning in a clear and concise manner in a single column, which is a feat in itself.

**Questions For Authors:**

1) To what extent your method can be applied to online learning?

2) The goal of the authors is to achieve a decoupled representation of rewards and costs. However, can't this goal be achieved by defining corresponding Q and value function for costs and rewards? These could then be used directly in their framework.

Also, please answers points (1), (2), and (4) in the Claims section.

**Relation To Broader Scientific Literature:**

The authors compare frequently to [Zheng et al., 2024], as one of the first examples of application of Hamilton-Jacobi reachability for safety analysis. However, the empirical results demonstrate the inability of their algorithm (FISOR) to achieve absolute safety guarantees.
It looks like the main innovation wrt [Zheng et al., 2024] is the decoupled representation, which leads to a superior experimental performance.

**Theoretical Claims:**

Proofs are provided in the appendix. I did not check all details, but the theoretical claims look correct.

---

> ### Author Rebuttal · Authors · 2025-03-31
>
> We sincerely acknowledge Reviewer cp6m for providing the insightful feedback. Please see our response to your concerns below.
> ## (Claim1) Motivation: How are safe offline datasets collected?
> In offline RL, datasets are typically collected from past human experiences. For example, thousands of car accidents occur every day. These unsafe driving records, combined with safe ones, can be used to build a safe offline RL dataset for training autonomous driving systems. Compared to allowing UGVs to explore the real world through an online trial-and-error manner, it is both safer and more efficient.
>
> ## (Claim2) How is it possible to decouple the representations according to $Q^{\*}$?
>
> When focusing solely on the reward $r$, the optimal value $Q_r^{*}$ is inherently independent of cost information $c$, as defined by its formulation. For instance, as shown in Figure 1 (page 1), all states in the figure share the same $Q_r^{\*}$ across all actions. Leveraging this property, we cluster the mid-layer representations of states with similar $Q_r^{\*}$ values across actions within the support, ensuring **these representations remain independent of cost-related information**.
> Likewise, the same principle applies when considering only the cost.
> For further intuition, please refer to the GIF in the supplementary material, which visualizes policies derived from the decoupled states.
>
> ## (Claim4) What does "generalization" refer to?
>
> We apologize for any confusion caused by the lack of explanation regarding this terminology. In machine learning, "generalization" usually refers to a model's ability to perform well on new, unseen data not included in its training set.
>
> In real-world decision-making, state spaces are often high-dimensional and infinite, making it impossible to cover all possible states during training phase.
> This is especially true in offline settings, where the agent is trained on a fixed dataset without exploration [1].
> The online testing performance of an offline-trained agent directly reflects the algorithm's generalization ability [1]. During deployment, the agent may encounter unfamiliar states and take suboptimal or unsafe actions. As shown in Table 1, our SDQC algorithm achieves SOTA performance in both rewards and costs.
>
> Moreover, our approach extends generalization to more complex scenarios, where the testing environment (containing more obstacles) differs significantly from the one in which the offline data was collected, as displayed in Figure 3 and Figure 9.
> (For online RL, generalization is usually evaluated in a similar way [2].)
>
> ## (Question1) Applying SDQC to online settings.
>
> We agree that SDQC has great potential for extension to online settings. However, the in-sample learning method (implicit-Q learning) used in our work is specifically designed for offline RL.
> Our primary contribution is the introduction of the concept of "decoupling the states for decision-making," which can be seamlessly integrated with many existing online RL methods that use representations like bisimulation. Unfortunately, due to time constraints, we were unable to verify this point during the rebuttal period.
>
> ## (Question2) Possibilities of abstracting representations from pre-trained value functions.
>
> If we understand correctly, the reviewer is asking whether it is possible to first train the value functions through in-sample learning and then extract the representations based on the converged value functions.
> We argue that this approach would make the training process more complex and prone to failure.
>
> Specifically, given any global observation $s$, we should firstly obtain its cost-related representation $z_h$ then make safety assessment on representations $z_h$ according to value function $V_h(z_h)$. This process requires learning two mappings: $\mathcal{S} \mapsto \mathcal{Z}_h \mapsto \mathcal{V}_h$.
> However, if we were to train the value functions first and subsequently extract the representations based on these pre-trained value functions, the task would become  more complex. This is because an additional mapping would be introduced: $\mathcal{S} \mapsto \mathcal{V}_h \mapsto \mathcal{Z}_h \mapsto \mathcal{V}_h$. In this case, the mapping between the **vector space** $\mathcal{Z}_h$ and the **scalar space** $\mathcal{V}_h$ would be **bijective**, which is extremely challenging for neural networks to learn. Furthermore, this approach would be susceptible to additional errors due to the approximation nature of neural networks.
>
> In contrast, our approach utilizes the neural network structure presented in Figure 5 (Appendix C.2, page 19). The representations are explicitly set as mid-layer outputs, and only the necessary mappings are learned.
>
> [1] Levine, Sergey, et al. Offline reinforcement learning: Tutorial, review, and perspectives on open problems. arXiv:2005.01643 (2020).
>
> [2] Agarwal, Rishabh, et al. Contrastive behavioral similarity embeddings for generalization in reinforcement learning. ICLR (2021).

---

### Decision · Program_Chairs · 2025-05-01

**Decision:**

Accept (poster)

**Comment:**

This work proposes a method for safe offline RL that decomposes states into reward- and cost-specific representations. They theoretically contrast their representation learning approach with bisimulation-metrics-based approaches, and prove that they still maintain optimality. The empirical evaluations are rather thorough, and appendix C is quite commendable for its efficacy in explaining the method.

While there are some concerns about the complexity of the method, the ablations provided suggest it is relatively robust to changes to them.

Most reviewers are supportive of acceptance, as am I.